# Diffusion Representation for Asymmetric Kernels via Magnetic Transform

**Mingzhen He**
Institute of Image Processing and Pattern Recognition
Shanghai Jiao Tong University, Shanghai, China
`mingzhen_he@sjtu.edu.cn`

**Fan He**
Department of Electrical Engineering (ESAT-STADIUS)
KU Leuven, Leuven, Belgium
`fan.he@esat.kuleuven.be`

**Ruikai Yang, Xiaolin Huang**
Institute of Image Processing and Pattern Recognition
Shanghai Jiao Tong University, Shanghai, China
`{ruikai.yang, xiaolinhuang}@sjtu.edu.cn`

## Abstract

As a nonlinear dimension reduction technique, the diffusion map (DM) has been widely used. In DM, kernels play an important role for capturing the nonlinear relationship of data. However, only symmetric kernels can be used now, which prevents the use of DM in directed graphs, trophic networks, and other real-world scenarios where the intrinsic and extrinsic geometries in data are asymmetric. A promising technique is the magnetic transform which converts an asymmetric matrix to a Hermitian one. However, we are facing essential problems, including how diffusion distance could be preserved and how divergence could be avoided during diffusion process. Via theoretical proof, we successfully establish a diffusion representation framework with the magnetic transform, named *MagDM*. The effectiveness and robustness for dealing data endowed with asymmetric proximity are demonstrated on three synthetic datasets and two trophic networks.

## 1 Introduction

The diffusion map (DM) [1, 2, 3] and its variations [4, 5, 6, 7] are typical dimension reduction (DR) techniques that has been widely used in the signal processing [8], computer vision [9], and textual network embedding [10]. DM is a powerful method that embeds high-dimensional data into a low-dimensional manifold while preserving diffusion distance, which helps describe meaningful geometric structures in data.

Kernels play a significant role in DM and strongly impact the quality of DR, as they allow for the identification of nonlinear relationships and underlying structures in data. In real-world scenarios such as directed graphs, trophic and social networks, the intrinsic and extrinsic geometries in data may be asymmetric, and asymmetric kernels [11, 12, 13, 14] are more suitable to depict asymmetric geometric structures. The advantages of integrating diffusion geometry and asymmetric kernels have been acknowledged by several researchers. [15] developed a framework to study function extension and approximation problems in the context of changing data and functions on directed graphs.

37th Conference on Neural Information Processing Systems (NeurIPS 2023).

However, the diffusion distance and diffusion maps for vertices on directed graphs were unclear. [16] proposed the diffusion map for asymmetric kernels using the singular value decomposition, while the diffusion distance was calculated using only left singular vectors and right singular vectors were neglected, resulting in the underutilization of asymmetric information.

The magnetic transform (MT) [17] is a promising technique that transforms an asymmetric matrix into a Hermitian one, making it suitable for working with asymmetry. MT has been successfully applied in DR techniques based on asymmetric adjacency matrices on directed graphs, such as Laplacian eigenmaps [18, 19]. Nevertheless, connecting DM and MT still faces essential problems. Previous works have analyzed MT based on finite adjacency matrices of directed graphs, while the studies of applying MT on asymmetric kernel functions, the corresponding integral operators, and diffusion geometry have not been explored, which obstructs the application of MT to DM. Additionally, MT transforms real-valued asymmetric kernels that describe the connectivity between data into complex-valued function spaces, raising the question of how to define the diffusion distance and diffusion maps in those complex-valued function spaces. It is also crucial to investigate how to prevent divergence during the diffusion process when we are working in complex field. To address these challenges, we propose a novel diffusion representation framework with the magnetic transform, named *MagDM*, that enables the use of asymmetric kernels.

The contributions of this paper are as follows:

- We develop a diffusion representation framework for data endowed with asymmetric kernels using the magnetic transform, where the corresponding diffusion distance and diffusion maps are defined.

- We explore integral operators of MT, whose compactness is established if the asymmetric kernels are Hilbert-Schmidt kernels, helping us determine the range of usable asymmetric kernels.

- We prove an important property that the spectral radius of the proposed integral operator is 1, which ensures that MagDM will not diverge during the diffusion process.

- We present experiments that demonstrate the effectiveness and robustness of the MagDM algorithm on three synthetic datasets and two real-world trophic networks.

## 2  Related works

Dimension reduction techniques [20, 21, 22] have been extensively studied over the past few decades. Previous research has focused mainly on symmetric and positive semi-definite proximity. However, in recent years, researchers have become aware of benefits of asymmetric proximity and have proposed several insightful theories and algorithms. These methods fall into one of the following families.

**Singular value decomposition.** [15] presented a polar decomposition for the construction of kernels to embed directed graphs into manifolds to study harmonic analysis of functions on directed graphs. However, the diffusion representation for asymmetric kernels under this framework was not discussed. [16] utilized the singular value decomposition (SVD) for asymmetric kernels, and defined the diffusion distance for asymmetric kernels. There was a theoretical shortcoming in which diffusion distance was calculated using only left singular vectors and right singular vectors were neglected, meaning that only out-going proximity was considered, which resulted in the underutilization of asymmetric information. Although SVD embeds asymmetric kernels as an inner product of two feature mappings, it does not directly reflect the directional information, i.e., the skew-symmetric part, which is more important in asymmetric scenarios, in the feature mappings.

**Magnetic Laplacian.** The magnetic Laplacian [17] was originally derived from a quantum mechanical Hamiltonian of a particle under magnetic flux. [18, 19, 23] utilized magnetic Laplacian to embed asymmetric proximity as a complex-valued feature mapping, where symmetric and skew-symmetric parts were successfully separated into modules and phases, respectively. Magnetic Laplacian transformed an asymmetric matrix to a normalized Hermitian matrix $H$ whose eigenvalues are contained within the interval $[-1, 1]$ [24]. In Laplacian-typed methods, the feature spaces turned to be the (numerical) null spaces of $I - H$ [25], in which only the positive part of the spectrum of $H$ was used. However, the negative part of the spectrum also contains meaningful information which was ignored in previous works [18, 19], resulting in an inadequate use of asymmetric information.

# 3 Diffusion representation and diffusion distance

In this section, we briefly review the diffusion map. Assume $(X, \mu)$ is a $\sigma$-finite measure space where $X \subset \mathbb{R}^M$ is a dataset and $\mu$ is an associated measure which is non-degenerate on $X$. Let $\mathcal{K} : X \times X \to \mathbb{R}_+$ be a non-negative symmetric kernel which measures the local connectivity between two samples $\boldsymbol{x}, \boldsymbol{y} \in X$. And the Markov kernel is defined as follows,

$$\rho(\boldsymbol{x}, \boldsymbol{y}) := \frac{\mathcal{K}(\boldsymbol{x}, \boldsymbol{y})}{\sqrt{\nu(\boldsymbol{x})}\sqrt{\nu(\boldsymbol{y})}} \,, \tag{1}$$

where $\nu(\boldsymbol{x})$ is the volume form defined as $\nu(\boldsymbol{x}) = \int_X \mathcal{K}(\boldsymbol{x}, \boldsymbol{y})\mathrm{d}\mu(\boldsymbol{y})$. Assuming that the volume form does not vanish and $\mathcal{K} \in L^2(X \times X, \mu \otimes \mu)$ is a square integrable measurable function, then the operator $A : L^2(X, \mu) \to L^2(X, \mu)$ given by $A(f)(\boldsymbol{x}) = \int_X \rho(\boldsymbol{x}, \boldsymbol{y})f(\boldsymbol{y})\mathrm{d}\mu(\boldsymbol{y})$ is compact and self-adjoint. We can write $\rho(\boldsymbol{x}, \boldsymbol{y}) = \sum_{n \in \mathbb{N}} \lambda_n \phi_n(\boldsymbol{x})\phi_n(\boldsymbol{y})$ according to the spectral theorem, where $\{\lambda_n, \phi_n\}$ is the spectral decomposition of the operator $A$. For any positive natural number $t \in \mathbb{N}_+$, the diffusion distance at time $t$ between two samples $\boldsymbol{x}, \boldsymbol{y}$ can be defined as follows,

$$D^t(\boldsymbol{x}, \boldsymbol{y}) := \|\rho^t(\boldsymbol{x}, \cdot) - \rho^t(\boldsymbol{y}, \cdot)\|_{L^2(X, \mu)} \,, \tag{2}$$

where $\rho^t$ is the kernel of the $t$ powers of the integral operator $A$. Note that the diffusion distance $D^t$ is an average over all paths in time $t$ connecting $\boldsymbol{x}$ to $\boldsymbol{y}$, resulting in a diffusion distance that is robust to noise perturbation. The diffusion distance can be reformulated as follows,

$$D^t(\boldsymbol{x}, \boldsymbol{y}) = \sqrt{\sum_{n \in \mathbb{N}} \lambda_n^{2t}\big(\phi_n(\boldsymbol{x}) - \phi_n(\boldsymbol{y})\big)^2} \,.$$

The expression above enables dimension reduction in diffusion geometry, which is also known as diffusion maps. The dataset with the kernel structure $\rho$ can be embedded into a lower dimensional space $\psi_k^t : X \to \mathbb{R}^k$ as follows,

$$\psi_k^t(\boldsymbol{x}) = \Big[\lambda_1^t \phi_1(\boldsymbol{x}), \quad \lambda_2^t \phi_2(\boldsymbol{x}), \quad \cdots, \quad \lambda_k^t \phi_k(\boldsymbol{x})\Big]^\top \,.$$

# 4 Diffusion representation for asymmetric kernels

The definition (2) of the diffusion distance assumes symmetry, making it unsuitable for use with asymmetric kernels. To overcome this limitation, we introduce the magnetic transform in this section. With this transformation, we propose the MagDM algorithm for asymmetric kernels.

## 4.1 Magnetic transform

Assume that we have a set of data $X \subset \mathbb{R}^M$, and a non-negative distribution $\mu$ defined on this set. Let $\mathcal{K} : X \times X \to \mathbb{R}_+$ be a non-negative kernel that measures the directed connectivity between samples in $X$. Note that $\mathcal{K}$ does not need to be symmetric. The classical diffusion distance (2) fails to measure asymmetric distance from $\boldsymbol{x}$ to $\boldsymbol{y}$ because it only uses half of asymmetric information, treating both samples as sources. To handle asymmetry, the magnetic transform is defined to transform an asymmetric kernel to a new kernel with conjugate symmetry.

Suppose an asymmetric kernel $\mathcal{K} : X \times X \to \mathbb{R}_+$ is a Hilbert-Schmidt kernel, $\mathcal{K} \in L^2(X \times X, \mu \otimes \mu)$. The Magnetic transform kernel $\mathcal{H}^{(q)}$ of $\mathcal{K}$ is defined by

$$\mathcal{H}^{(q)}(\boldsymbol{x}, \boldsymbol{y}) = \frac{\mathcal{K}(\boldsymbol{x}, \boldsymbol{y}) + \mathcal{K}(\boldsymbol{y}, \boldsymbol{x})}{2} \exp\big(\mathrm{i}2\pi q \cdot \big(\mathcal{K}(\boldsymbol{y}, \boldsymbol{x}) - \mathcal{K}(\boldsymbol{x}, \boldsymbol{y})\big)\big) \,, \tag{3}$$

where $q \in \mathbb{R}^+$ is a scaling parameter and $\mathrm{i} = \sqrt{-1}$. The kernel $\mathcal{H}^{(q)}$ is conjugate symmetric, meaning that $\mathcal{H}^{(q)}(\boldsymbol{x}, \boldsymbol{y}) = \overline{\mathcal{H}^{(q)}(\boldsymbol{y}, \boldsymbol{x})}$. The modulus and phase terms of $\mathcal{H}^{(q)}$ capture the symmetric and skew-symmetric parts of information in $\mathcal{K}$, respectively.

**Proposition 1.** *Suppose a real-valued asymmetric kernel $\mathcal{K} : X \times X \to \mathbb{R}_+$ is a Hilbert-Schmidt kernel $\mathcal{K} \in L^2(X \times X, \mu \otimes \mu)$ where $(X, \mu)$ is a $\sigma$-finite measure space and $X \subset \mathbb{R}^M$. Then, the magnetic transform kernel $\mathcal{H}^{(q)} : X \times X \to \mathbb{C}$ is a Hilbert-Schmidt kernel.*

The proof is shown in Appendix A. The aforementioned proposition allows us to define a Hilber-Schmidt kernel $\mathcal{H}^{(q)}$, which enables us to identify the range of usable asymmetric kernels. Then, we can define the corresponding Hilber-Schmidt operator that is also a compact operator [26] in the diffusion process.

## 4.2 Diffusion maps and diffusion distance

The Markov kernel $\rho(\boldsymbol{x}, \boldsymbol{y}, q) \in L^2(X \times X, \mu \otimes \mu)$ for the magnetic transform kernel $\mathcal{H}^{(q)}$ is defined by

$$\rho(\boldsymbol{x}, \boldsymbol{y}, q) = \frac{\mathcal{H}^{(q)}(\boldsymbol{x}, \boldsymbol{y})}{\sqrt{m(\boldsymbol{x})}\sqrt{m(\boldsymbol{y})}}, \tag{4}$$

where $m(\boldsymbol{x}) = \frac{1}{2}\int_X \left(\mathcal{K}(\boldsymbol{x}, \boldsymbol{y}) + \mathcal{K}(\boldsymbol{y}, \boldsymbol{x})\right)\mathrm{d}\mu(\boldsymbol{y})$ is the volume form of the symmetric term of $\mathcal{K}$. Assuming that $m(\boldsymbol{x})$ does not vanish on $X$, the operator $T^{(q)} : L^2(X, \mu) \to L^2(X, \mu)$ given by

$$T^{(q)}(f)(\boldsymbol{x}) = \int_X \rho(\boldsymbol{x}, \boldsymbol{y}, q)f(\boldsymbol{y})\mathrm{d}\mu(\boldsymbol{y}) \tag{5}$$

is compact and self-adjoint. We can decompose $\rho(\boldsymbol{x}, \boldsymbol{y}, q) = \sum_{n \in \mathbb{N}} \lambda_n^{(q)}\phi_n^{(q)}(\boldsymbol{x})\phi_n^{(q)}(\boldsymbol{y})$ according to the spectral theorem, where $\{\lambda_n^{(q)}, \phi_n^{(q)}\}$ is the spectral decomposition of the operator $T^{(q)}$. Next, we analyze the spectrum of the integral operator (5).

**Proposition 2.** *Given an asymmetric kernel satisfying Proposition 1, the eigenvalues $\{\lambda_i^{(q)}\}$ of the Hilbert-Schmidt operator defined by (5) are real and bounded by $\lambda_i^{(q)} \in [-1, 1]$.*

The proof is provided in Appendix B. Because the operator is self-adjoint, its eigenvalues are real numbers and eigenfunctions are complex functions. According Proposition 2, as diffusion time $t \in \mathbb{N}$ increases, small eigenvalues converge to zero, which avoids divergence of MagDM during diffusion process. For any time $t$, MagDM defines the diffusion distance for the asymmetric kernel by

$$D^t(\boldsymbol{x}, \boldsymbol{y}) := \|\rho^t(\boldsymbol{x}, \cdot, q) - \rho^t(\boldsymbol{y}, \cdot, q)\|_{L^2(X, \mu)}. \tag{6}$$

Running along the diffusion process for asymmetric kernels is equivalent to computing the powers of the operator $T^{(q)}$. The diffusion distance (6) is a functional weighted $L^2$ distance between complex-valued proximities of two samples, thus it is also robust to noise perturbation. Both symmetry and skew-symmetry are embedded into the diffusion distance, which reflects the asymmetric geometry in data. (6) can be rewritten as the following,

$$D^t(\boldsymbol{x}, \boldsymbol{y}) = \sqrt{\sum_{n \in \mathbb{N}} \left(\lambda_n^{(q)}\right)^{2t}\left|\phi_n^{(q)}(\boldsymbol{x}) - \phi_n^{(q)}(\boldsymbol{y})\right|^2}. \tag{7}$$

Denote the eigenvalues in descending order as: $1 \geq |\lambda_1^{(q)}| \geq |\lambda_2^{(q)}| \geq \cdots \geq 0$. $D^t(\boldsymbol{x}, \boldsymbol{y})$ can be approximated by $D_\delta^t(\boldsymbol{x}, \boldsymbol{y})$ with a preset accuracy $\delta \in (0, 1)$ and finite terms as follows,

$$D_\delta^t(\boldsymbol{x}, \boldsymbol{y}) = \sqrt{\sum_{n \leq s(\delta, t)} \left(\lambda_n^{(q)}\right)^{2t}\left|\phi_n^{(q)}(\boldsymbol{x}) - \phi_n^{(q)}(\boldsymbol{y})\right|^2},$$

where $s(\delta, t) = \max\{n \in \mathbb{N}_+ : |\lambda_n^{(q)}|^t > \delta|\lambda_1^{(q)}|^t\}$. We define the diffusion map for asymmetric kernels which embeds data endowed with asymmetric kernels in a lower dimensional space as follows,

$$\psi^{t,(q)}(\boldsymbol{x}) = \left[\left(\lambda_1^{(q)}\right)^t\phi_1^{(q)}(\boldsymbol{x}), \quad \left(\lambda_2^{(q)}\right)^t\phi_2^{(q)}(\boldsymbol{x}), \quad \cdots, \quad \left(\lambda_{s(\delta,t)}^{(q)}\right)^t\phi_{s(\delta,t)}^{(q)}(\boldsymbol{x})\right]^\top. \tag{8}$$

**Proposition 3.** *The diffusion map $\psi^{t,(q)}$ embeds the data endowed with an asymmetric kernel $\mathcal{K}$ into a lower dimensional space $\mathbb{C}^{s(\delta,t)}$, then the Euclidean distance in this space is equal to the diffusion distance $D_\delta^t(\boldsymbol{x}, \boldsymbol{y})$,*

$$\|\psi^{t,(q)}(\boldsymbol{x}) - \psi^{t,(q)}(\boldsymbol{y})\|_2 = D_\delta^t(\boldsymbol{x}, \boldsymbol{y}).$$

When the fed kernel function is symmetric, the proposed method reverts back to the classical diffusion map. This is because $\exp\left(\mathrm{i}2\pi q \cdot \left(\mathcal{K}(\boldsymbol{x}, \boldsymbol{y}) - \mathcal{K}(\boldsymbol{y}, \boldsymbol{x})\right)\right)$ becomes $\exp\left(\mathrm{i}2\pi q \cdot 0\right) = 1$, and $\mathcal{H}^{(q)}(\boldsymbol{x}, \boldsymbol{y}) = \mathcal{K}(\boldsymbol{x}, \boldsymbol{y})$. The implementation details of this algorithm can be found in Appendix D.

### 4.3 Selection of the scaling parameter q.

The scaling parameter $q$ is a key parameter for embedding the skew-symmetric part of an asymmetric kernel, which directly affects the representation ability. The selection of $q$ has been previously discussed for unweighted directed graphs [18, 23] and in the context of graph signal processing [27]. However, the selection of $q$ for asymmetric kernel functions has not been addressed so far. Here, we propose a simple and practical method for selecting $q$ for asymmetric kernels. In (3), the skew-symmetry is embedded in the phase, which is a periodic function controlled by $q$. To select a suitable value for $q$, we recommend selecting a period that covers the extent of the skew-symmetric component, which means that $\overline{a}$ should be less than $\pi$ where $\overline{a} := \sup_{\boldsymbol{x},\boldsymbol{y}\in X} |\mathcal{K}(\boldsymbol{x},\boldsymbol{y}) - \mathcal{K}(\boldsymbol{y},\boldsymbol{x})|$, then we have $q < 0.5/\overline{a}$. In practice, we have access to the Gram matrix $\boldsymbol{K} \in \mathbb{R}^{N\times N}$ where $N$ is the size of the data. Then $\overline{a}$ is given by $\overline{a} = \max_{i,j=1}^{N} |\boldsymbol{K}_{ij} - \boldsymbol{K}_{ji}|$, and the selection of $q$ is

$$0 \leq q < \frac{1}{2\max_{i,j=1}^{N} |\boldsymbol{K}_{ij} - \boldsymbol{K}_{ji}|} .$$

## 5 Experiments

In this section, we demonstrate the capability of the MagDM method to extract asymmetric information on three synthetic and two real-world trophic datasets. Additionally, we compare our method with five other methods: diffusion maps (DM) [1], kernel principal component analysis (KPCA) [28], asymmetric diffusion maps (ADM) [16], magnetic eigenmaps (ME) [18], and Markov normalized magnetic eigenmaps (MME) [19]. The experiments were conducted using MATLAB on a PC with an Intel i7-10700K CPU (3.8GHz) and 32GB of memory. Codes are available at `https://github.com/AlexHe123/MagDM`

### 5.1 Artificial networks

We begin by assessing the ability to extract asymmetric information. In this experiment, we create an artificial network following [29]. The network consists of three groups (A, B, and C), and we establish two types of flows (forward and backward) between them, which helps us construct an adjacency matrix to represent connections among nodes. Each node has a $50\%$ chance of being connected to a node in its own group. Furthermore, each node has a $50\%$ chance of being connected to a node in another group in the direction of the forward flow (i.e., interconnections from A to B, from B to C, and from C to A). Additionally, there is another type of interconnection in the direction of the backward flow with a probability of $P$. By adjusting the backward probability $P$, we can control the running flow. An illustration of this artificial network is shown in Appendix F.1.

In this experiment, we choose 5 probabilities for the backward running flow, $P \in \{0, 0.2, 0.5, 0.8, 1\}$. As $P$ increases, the level of asymmetric information first decreases and then increases. Due to the fact that phases of eigenvectors contain most of asymmetric information [18, 19], Fig. 1 reports phases of the ME, MME and MagDM methods, and shows their capability of extracting asymmetric information. All the three methods distinguish three groups when the probability of backward flow $P = 0$. As we increase the value of $P$, the level of asymmetric information decreases. This is reflected in the fact that the distance between samples in the same group becomes larger, as seen in the second row of Fig. 1. When $P = 0.5$, the contribution of the forward and backward flows to adjacency connections is statistically the same, resulting in the disappearance of asymmetric information and the mixing of samples from different groups. As $P$ continues to increase, asymmetric information gradually strengthens again, forming a scenario where the backward flow is stronger than the forward flow. The last two rows of Figure 1 demonstrate that the MagDM algorithm effectively distinguishes between the three groups, whereas the ME and MME algorithms are currently unable to differentiate between them, which illustrates the stronger robustness of our method.

The running flow of the second artificial network comprises four groups (A, B, C, and D) following [30]. The visualization of this directed graph can be seen in Appendix F.2. The structure of the flow is apparent, with Groups A and D serving as out-come and in-come flows, while Groups B and C function as intermediaries. For instance, Group A predominantly generates out-coming links with a high probability (strong flow), and only a few in-coming links from Group D are produced by weak flow with a low probability. Besides, each node has a high probability of being connected to a node in the same group.

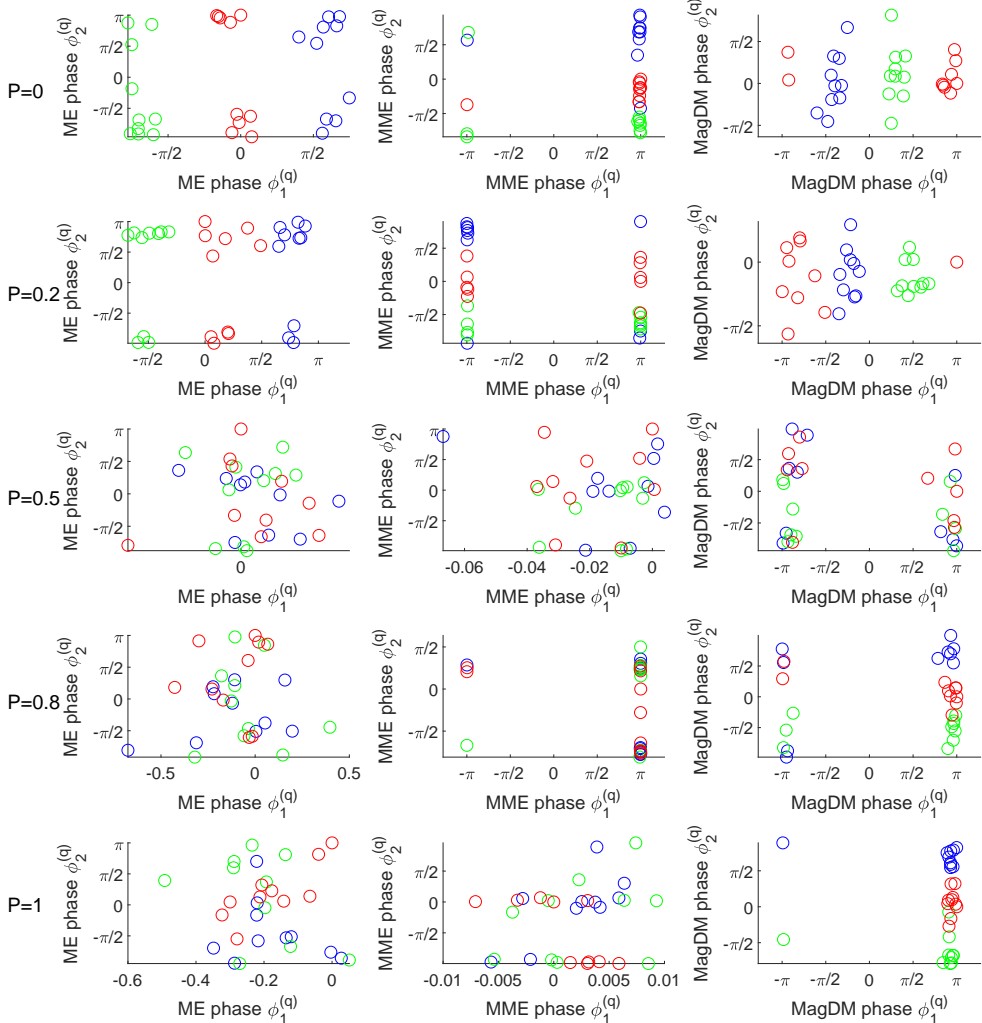

Figure 1: The capability of three methods (left: ME, middle: MME, right: MagDM) to extract asymmetric information on the three groups colored by red, green, and blue, with $q = 1/4$. Asymmetry in the data is generated by the asymmetric interconnection among the groups, and phases of the three methods are reported to distinguish them. The forward flow probability is fixed at $0.5$, while the backward flow probability is chosen from the set $\{0, 0.2, 0.5, 0.8, 1\}$.

Fig. 2 presents the visualization results of six different methods. The DM and KPCA methods yield poor clustering performance as they do not consider asymmetric information. In contrast, the ADM method distinguishes four groups, although it does not illustrate directed information. The ME method distinguishes groups along the phase axes, but it is unable to differentiate between the four groups when considering the real and imaginary parts of the second eigenvector. The MME method has an overlap between Groups A and B along the phase axes, and the second eigenvector embeds Group A within the other groups, making it difficult to distinguish it from the rest. MagDM produces a compact and almost linearly separable clustering not only along the phase axes but also in the first and second eigenvectors, as shown in the bottom row of Fig. 2.

## 5.2 Synthetic data using an asymmetric kernel

Next, we assume that the dataset is a set of 300 points distributed along the Möbius strip. The parametric form of the Möbius strip is defined by

$$x(u,v) = \left(1 + \frac{v}{2}\cos\frac{u}{2}\right)\cos u, \quad y(u,v) = \left(1 + \frac{v}{2}\cos\frac{u}{2}\right)\sin u, \quad z(u,v) = \frac{v}{2}\sin\frac{u}{2},$$

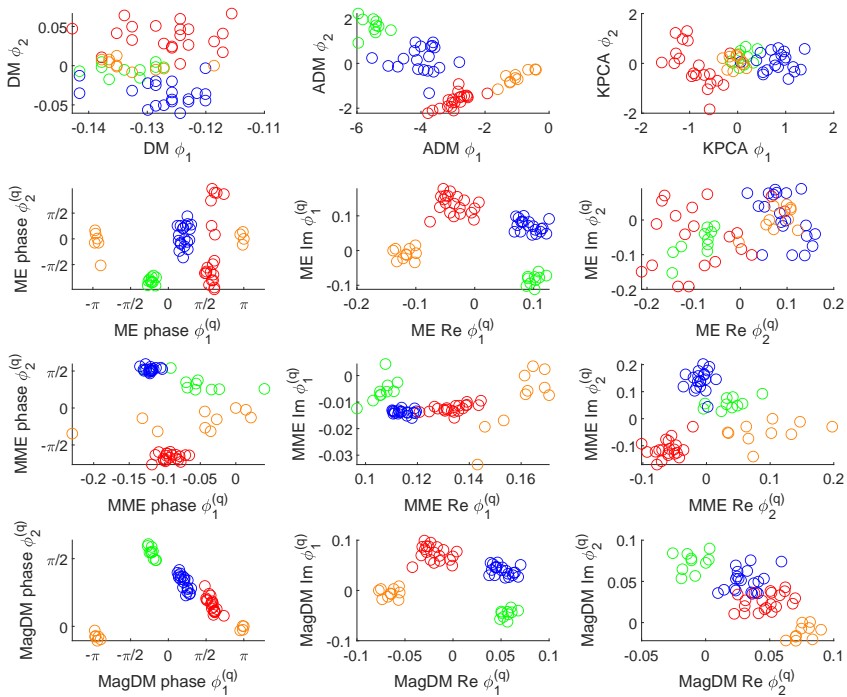

Figure 2: The capability of six methods to extract asymmetric geometry on the four groups colored by red, green, blue, and orange, with $q = 1/3$. The first row shows the results of DM, ADM, and KPCA, while the remaining rows show the results of ME, MME, and MagDM. For better illustration, the real and imaginary parts of first two non-trivial eigenvectors are reported, in addition to the phases.

where $0 \leq u \leq 2\pi$ and $-0.5 \leq v \leq 0.5$. The dataset is with a color drift in the counterclockwise direction on the x-y plane, please see it in Appendix F.3. We define an asymmetric kernel as follows,

$$
\mathcal{K}(\boldsymbol{x}, \boldsymbol{y}) = \begin{cases} \max\{\rho - \|\boldsymbol{x} - \boldsymbol{y}\|_1, 0\}, & \angle \boldsymbol{x}\boldsymbol{y} \geq 0 \\ \max\{0.2\rho - \|\boldsymbol{x} - \boldsymbol{y}\|_1, 0\}, & \angle \boldsymbol{x}\boldsymbol{y} < 0 \end{cases}, \tag{9}
$$

where $\angle \boldsymbol{x}\boldsymbol{y}$ is the angle between $\boldsymbol{x}, \boldsymbol{y}$ on the x-y plane. The asymmetric kernel is composed of two truncated $\ell_1$ kernels [31], which measures asymmetric similarity based on angles between $\boldsymbol{x}$ and $\boldsymbol{y}$.

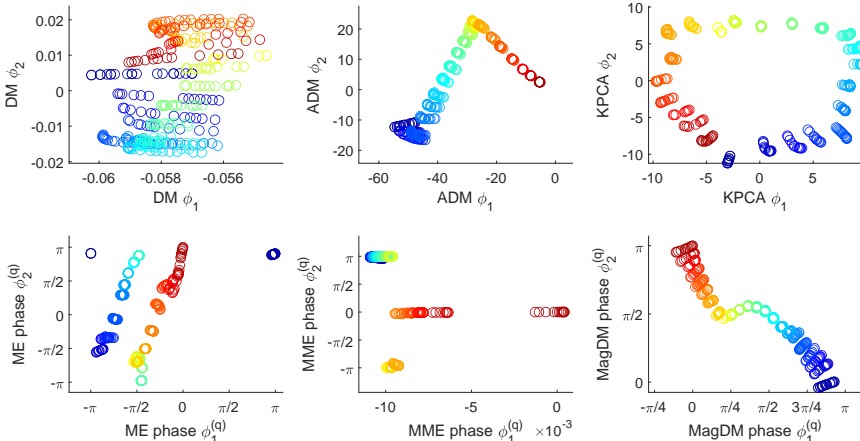

Figure 3: Dimension reduction results of six methods (DM, ADM, KPCA, ME, MME, and MagDM) on the Möbius strip which is endowed with the asymmetric kernel (9) and colored by the angle. $\rho = 5$ and $q = 0.09$.

The dimension reduction results are illustrated in Fig. 3. DM and KPCA fail to distinguish the asymmetric geometry, as they only learn the symmetric part of (9). Although the ADM method contains the color drift after reducing the dimension of data, there is still a sharp turn on the green part of the data, which is not desired. The ME method distinguishes the color drift only on the phase of the first eigenvector, while the MME method fails to provide clear directed information. Notably, the MagDM method provides a clear and smooth result along both two phase axes.

## 5.3 Trophic networks

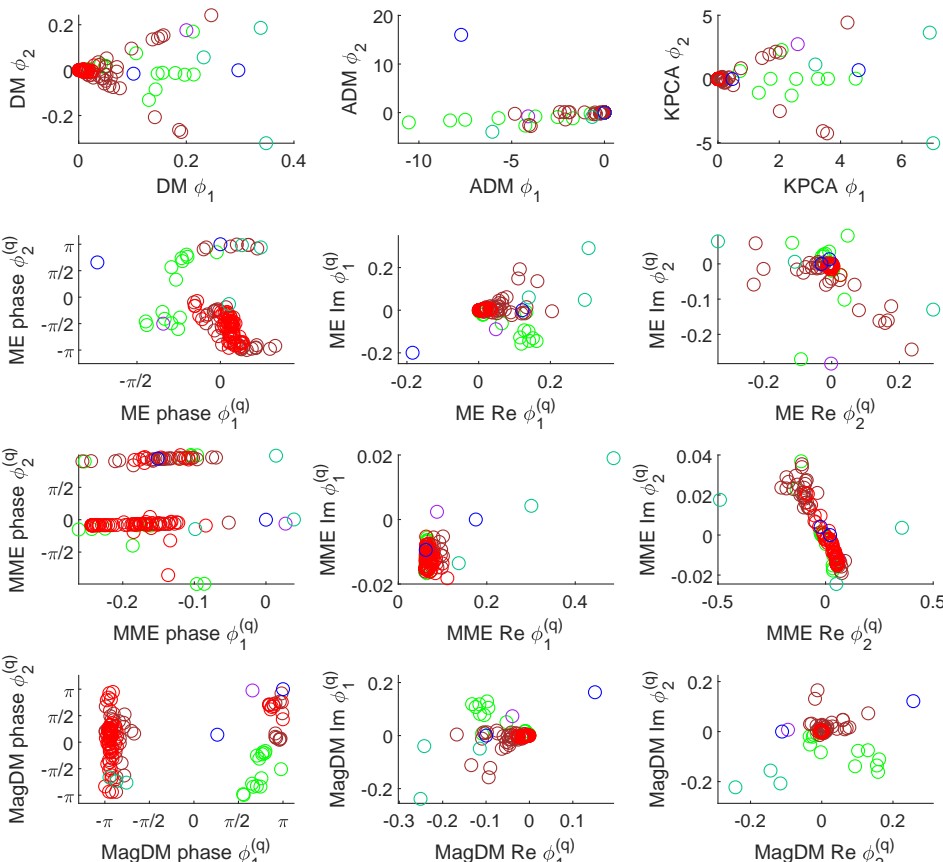

Figure 4: The dimension reduction of the Florida network is shown using six methods (DM, ADM, KPCA, ME, MME, and MagDM), with $q = 0.045$. Florida is composed of four categories consisting of different elements: Producers (green nodes), Consumers (red and brown nodes), Organic Matters (blue and turquoise blue nodes), and one Decomposer (purple node). The low-dimensional embeddings using the different methods are visually compared.

To further illustrate the concept, we have chosen two specific real-world trophic networks from the Pajek datasets[1]: the Mondego network [32], which records trophic exchanges at the Mondego estuary, and the Florida network [33], which records trophic exchanges in Florida Bay during the wet season.

Based on the roles of the nodes in ecosystems, we classify them into different categories, as depicted in Appendix F.4. The different categories in the trophic networks consist of Producers, which generate their own food through photosynthesis or chemosynthesis (green nodes); Consumers, consisting of both low-level consumers (brown nodes) and high-level consumers (red nodes) that feed on other organisms for energy; and Organic Matters (blue and turquoise blue nodes). As for the Florida network, there is also a Decomposer that break down dead or decaying organic matter (purple node). The adjacency matrix $W$ denotes the intensity of trophic exchanges and the asymmetric kernel matrix

---

[1]http://vlado.fmf.uni-lj.si/pub/networks/data/bio/foodweb/foodweb.htm

for the trophic networks are defined by,

$$\boldsymbol{K}_{ij} = \log_2(\boldsymbol{W}_{ij} + 1).$$

The dimension reduction results for two trophic networks are presented in Figs. 4 and 5. For the Florida network, MagDM outperforms other methods, as shown in Fig. 4. All DM, ADM, and KPCA methods fail to differentiate between different categories. On the phases results of the remaining three methods, ME and MME are unable to extract clear asymmetric information, while MagDM almost distinguishes between producers and consumers. For better illustration, the real and imaginary parts of the first two non-trivial eigenvectors of ME, MME, and MagDM are visually compared. It is observed that all three methods fail to distinguish categories on the first eigenvector axis, but on the second eigenvector axis, MagDM not only differentiates the categories but also tightly clusters high-level consumers together, resulting in a clear classification structure. Similar results are obtained for the Mondego network. These findings demonstrate that MagDM is a promising approach for dealing with asymmetric proximity in trophic networks.

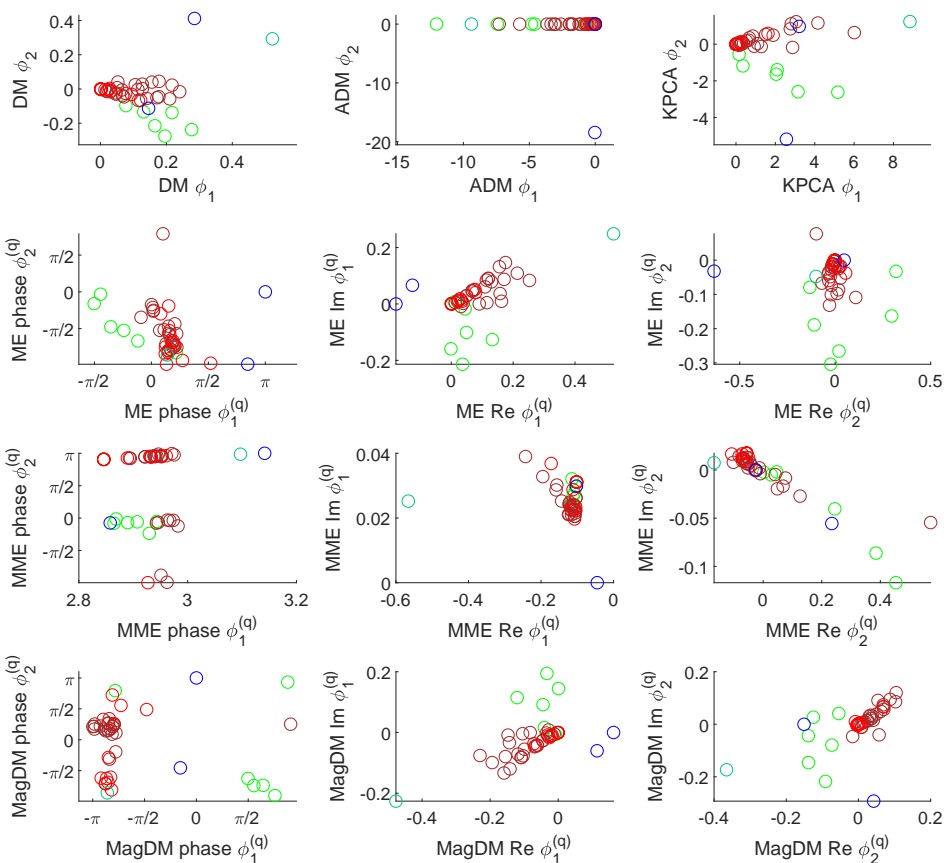

Figure 5: The dimension reduction of the Mondego network is shown using six methods (DM, ADM, KPCA, ME, MME, and MagDM), with $q = 0.04$. Mondego is composed of three categories consisting of different elements: Producers (green nodes), Consumers (red and brown nodes), and Organic Matters (blue and turquoise blue nodes). The low-dimensional embeddings using the different methods are visually compared.

# 6  Conclusion

In this paper, we have presented a novel framework named MagDM for representing data endowed with asymmetric kernels using the magnetic transform within the context of diffusion maps. We introduced the integral operator with the proposed magnetic transform kernel and investigated several

of its properties, including compactness, spectral decomposition, and spectral radius. Moreover, we proposed a diffusion distance that embeds asymmetric kernels into the diffusion map in complex-valued function spaces, allowing the identification and analysis of asymmetric geometries and patterns in data. This could lead to new insights into the structure and properties of asymmetric kernels, potentially leading to new applications or algorithmic improvements. Experiments demonstrated the effectiveness and robustness of MagDM on three synthetic datasets and two trophic networks.

There are several interesting avenues for future work in the study of asymmetric kernels. In terms of methodology, exploring variants of other dimension reduction schemes for asymmetric kernels such as MDS [34] and Isomap [35] would be appealing. In terms of theory, studying geometric harmonics of functions with magnetic transform kernels could provide a deeper understanding of the intrinsic and extrinsic geometries in data. In terms of applications, using MagDM as feature mappings for classification and regression tasks could be particularly useful for problems involving asymmetric proximity, such as social networks and commodity trading between economies.

**Broader impacts.** The proposed MagDM method described in this paper is not expected to have significantly different impacts on society compared to other dimension reduction techniques.

## Acknowledgments

This work was partially supported by National Natural Science Foundation of China (62376155), Research Program of Shanghai Municipal Science and Technology Committee (22511105600), and Shanghai Municipal Science and Technology Major Project (2021SHZDZX0102).

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

## A    The proof of Proposition 1

*Proof.* Because $\mathcal{K}$ is a Hilbert-Schmidt kernel, we have $\iint_{X \times X} |\mathcal{K}(\boldsymbol{x}, \boldsymbol{y})|^2 \mathrm{d}\mu(\boldsymbol{x})\mathrm{d}\mu(\boldsymbol{y}) < \infty$. Then,

$$
\begin{aligned}
\iint_{X \times X} |\mathcal{H}^{(q)}(\boldsymbol{x}, \boldsymbol{y})|^2 \mathrm{d}\mu(\boldsymbol{x})\mathrm{d}\mu(\boldsymbol{y}) &= \int_X \int_X |\frac{\mathcal{K}(\boldsymbol{x}, \boldsymbol{y}) + \mathcal{K}(\boldsymbol{y}, \boldsymbol{x})}{2}|^2 \mathrm{d}\mu(\boldsymbol{x})\mathrm{d}\mu(\boldsymbol{y}) \\
&\leq \int_X \int_X \frac{1}{2} \left( |\mathcal{K}(\boldsymbol{x}, \boldsymbol{y})|^2 + |\mathcal{K}(\boldsymbol{y}, \boldsymbol{x})|^2 \right) \mathrm{d}\mu(\boldsymbol{x})\mathrm{d}\mu(\boldsymbol{y}) \\
&= \int_X \int_X |\mathcal{K}(\boldsymbol{x}, \boldsymbol{y})|^2 \mathrm{d}\mu(\boldsymbol{x})\mathrm{d}\mu(\boldsymbol{y}) < \infty \, .
\end{aligned}
$$

It holds that the symbol " $\leq$ " in the third row of the above equation because of the Cauchy-Schwarz inequality. The Cauchy-Schwarz inequality states that $\forall a_i, b_i \in \mathbb{R}, i = 1, \cdots, k$, then $\left( \sum_{i=1}^k a_i b_i \right)^2 \leq \left( \sum_{i=1}^k a_i^2 \right) \left( \sum_{i=1}^k b_i^2 \right)$. Then, we have $|\frac{\mathcal{K}(\boldsymbol{x},\boldsymbol{y})}{2} + \frac{\mathcal{K}(\boldsymbol{y},\boldsymbol{x})}{2}|^2 \leq (\frac{1}{4} + \frac{1}{4}) \cdot \left( |\mathcal{K}(\boldsymbol{x}, \boldsymbol{y})|^2 + |\mathcal{K}(\boldsymbol{y}, \boldsymbol{x})|^2 \right)$, and measure $\mu$ is non-negative. Thus, symbol " $\leq$ " in the third row holds. We finish the proof. $\square$

## B    The proof of Proposition 2

*Proof.* The associated kernel $\mu(\boldsymbol{x}, \boldsymbol{y}, q)$ of the operator (5) is a Hermitian kernel, i.e., $\mu(\boldsymbol{x}, \boldsymbol{y}, q) = \overline{\mu(\boldsymbol{y}, \boldsymbol{x}, q)}$, then the eigenvalues of (5) are real and the spectral radius is equal to the operator norm $R(\rho) = \|T^{(q)}\|$ [36]. Let $f \in L^2(X, \mu)$, we have

$$
\begin{aligned}
\langle T^{(q)} f, f \rangle &= \int_X \int_X \rho(\boldsymbol{x}, \boldsymbol{y}, q) f(\boldsymbol{y})\overline{f(\boldsymbol{x})} \mathrm{d}\mu(\boldsymbol{x})\mathrm{d}\mu(\boldsymbol{y}) \\
&= \int_X \int_X \mathcal{H}^{(q)}(\boldsymbol{x}, \boldsymbol{y}) \frac{f(\boldsymbol{y})}{\sqrt{m(\boldsymbol{y})}} \frac{\overline{f(\boldsymbol{x})}}{\sqrt{m(\boldsymbol{x})}} \mathrm{d}\mu(\boldsymbol{x})\mathrm{d}\mu(\boldsymbol{y}) \, .
\end{aligned}
$$

If we apply the Cauchy-Schwarz inequality as follows,

$$
\begin{aligned}
\left| \int_X \mathcal{H}^{(q)}(\boldsymbol{x}, \boldsymbol{y}) \frac{f(\boldsymbol{y})}{\sqrt{m(\boldsymbol{y})}} \mathrm{d}\mu(\boldsymbol{y}) \right| &\leq \left( \int_X |\mathcal{H}^{(q)}(\boldsymbol{x}, \boldsymbol{y})| \mathrm{d}\mu(\boldsymbol{y}) \right)^{\frac{1}{2}} \left( \int_X |\mathcal{H}^{(q)}(\boldsymbol{x}, \boldsymbol{y})^{\frac{1}{2}} \frac{f(\boldsymbol{y})}{\sqrt{m(\boldsymbol{y})}}|^2 \mathrm{d}\mu(\boldsymbol{y}) \right)^{\frac{1}{2}} \\
&\leq \left( \int_X |\mathcal{H}^{(q)}(\boldsymbol{x}, \boldsymbol{y})| \mathrm{d}\mu(\boldsymbol{y}) \right)^{\frac{1}{2}} \left( \int_X |\mathcal{H}^{(q)}(\boldsymbol{x}, \boldsymbol{y})| \frac{|f(\boldsymbol{y})|^2}{m(\boldsymbol{y})} \mathrm{d}\mu(\boldsymbol{y}) \right)^{\frac{1}{2}} \\
&= \left( \int_X S(\boldsymbol{x}, \boldsymbol{y}) \mathrm{d}\mu(\boldsymbol{y}) \right)^{\frac{1}{2}} \left( \int_X S(\boldsymbol{x}, \boldsymbol{y}) \frac{|f(\boldsymbol{y})|^2}{m(\boldsymbol{y})} \mathrm{d}\mu(\boldsymbol{y}) \right)^{\frac{1}{2}} \, ,
\end{aligned}
$$

where $S(\boldsymbol{x}, \boldsymbol{y}) = |\mathcal{H}^{(q)}(\boldsymbol{x}, \boldsymbol{y})| = \frac{1}{2}|\mathcal{K}(\boldsymbol{x}, \boldsymbol{y}) + \mathcal{K}(\boldsymbol{y}, \boldsymbol{x})|$. The asymmetric kernel $\mathcal{K}$ is non-negative, thus $S(\boldsymbol{x}, \boldsymbol{y}) = \frac{1}{2}\left( \mathcal{K}(\boldsymbol{x}, \boldsymbol{y}) + \mathcal{K}(\boldsymbol{y}, \boldsymbol{x}) \right)$. Then we have

$$
\begin{aligned}
\left| \int_X \mathcal{H}^{(q)}(\boldsymbol{x}, \boldsymbol{y}) \frac{f(\boldsymbol{y})}{\sqrt{m(\boldsymbol{y})}} \mathrm{d}\mu(\boldsymbol{y}) \right| &\leq \left( \int_X S(\boldsymbol{x}, \boldsymbol{y}) \mathrm{d}\mu(\boldsymbol{y}) \right)^{\frac{1}{2}} \left( \int_X S(\boldsymbol{x}, \boldsymbol{y}) \frac{|f(\boldsymbol{y})|^2}{m(\boldsymbol{y})} \mathrm{d}\mu(\boldsymbol{y}) \right)^{\frac{1}{2}} \\
&= \sqrt{m(\boldsymbol{x})} \left( \int_X S(\boldsymbol{x}, \boldsymbol{y}) \frac{|f(\boldsymbol{y})|^2}{m(\boldsymbol{y})} \mathrm{d}\mu(\boldsymbol{y}) \right)^{\frac{1}{2}} \, .
\end{aligned}
$$

Consequently,

$$
\langle T^{(q)} f, f \rangle \leq \int_X |f(\boldsymbol{x})| \left( \int_X S(\boldsymbol{x}, \boldsymbol{y}) \frac{|f(\boldsymbol{y})|^2}{m(\boldsymbol{y})} \mathrm{d}\mu(\boldsymbol{y}) \right)^{\frac{1}{2}} \mathrm{d}\mu(\boldsymbol{x}) .
$$

We can apply the Cauchy-Schwarz inequality again,

$$\langle T^{(q)}f, f\rangle \le \|f\| \left( \int_X \int_X S(\boldsymbol{x}, \boldsymbol{y}) \frac{|f(\boldsymbol{y})|^2}{m(\boldsymbol{y})} \mathrm{d}\mu(\boldsymbol{y}) \mathrm{d}\mu(\boldsymbol{x}) \right)^{\frac{1}{2}}$$

$$= \|f\| \left( \int_X \frac{|f(\boldsymbol{y})|^2}{m(\boldsymbol{y})} \left( \int_X S(\boldsymbol{x}, \boldsymbol{y}) \mathrm{d}\mu(\boldsymbol{x}) \right) \mathrm{d}\mu(\boldsymbol{y}) \right)^{\frac{1}{2}}$$

$$= \|f\| \left( \int_X \frac{|f(\boldsymbol{y})|^2}{m(\boldsymbol{y})} m(\boldsymbol{y}) \mathrm{d}\mu(\boldsymbol{y}) \right)^{\frac{1}{2}} = \|f\|^2 \,.$$

It can be noticed that the operator norm $\|T^{(q)}\| = 1$. Therefore, the spectral radius $R(\rho) = 1$. We finish the proof.

$\square$

## C  Selection of the scaling parameter q.

As illustrated in Fig. Ape.1, we suggest choosing a period that encompasses the range of the skew-symmetric component, i.e., $\overline{a}$ should be less than $\pi$, where $\overline{a}$ is defined as follows,

$$\overline{a} := \sup_{\boldsymbol{x}, \boldsymbol{y} \in X} |\mathcal{K}(\boldsymbol{x}, \boldsymbol{y}) - \mathcal{K}(\boldsymbol{y}, \boldsymbol{x})| \,,$$

and the period of the phase function is $T = \frac{2\pi}{2\pi q} = 1/q$. Thus, we have

$$T = \frac{1}{q} > 2\overline{a} \quad \Rightarrow \quad q < \frac{1}{2\overline{a}} \,.$$

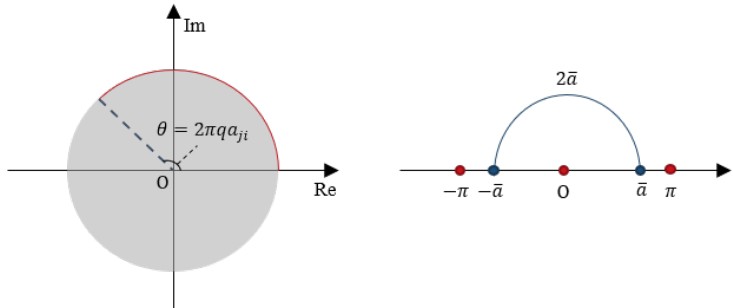

Figure Ape.1: A simple illustration for selecting $q$.

## D  Algorithm

To apply MagDM, we require a dataset and an asymmetric kernel or an asymmetric Gram matrix, as well as the scaling parameter $q$ and the desired accuracy. Algorithm 1 outlines the MagDM procedure.

---

**Algorithm 1** MagDM for asymmetric kernels

---

**Input:** The Gram matrix $\boldsymbol{K}$ of dataset $X$ endowed with an asymmetric kernel $\mathcal{K}$, the scaling parameter $q$ and a preset accuracy $\delta$.

**Output:** The diffusion map $\psi^{t,(q)}$ of $X$.

1: Calculate the Hermitian Gram matrix $\boldsymbol{H}$ of the asymmetric Gram matrix $\boldsymbol{K}$ by (3) and (4).
2: Calculate the $t$-powers kernel matrix $\boldsymbol{H}^t$.
3: Run eigen-decomposition of $\boldsymbol{H}^t$ and denote its eigen-system as $\{\lambda_n^{(q)}, \phi_n^{(q)}\}$.
4: $s(\delta, t) \leftarrow \max\{n \in \mathbb{N} : |\lambda_n^{(q)}| > \delta|\lambda_1^{(q)}|\}$.
5: Return the diffusion map $\psi^{t,(q)}$ by (8).

---

**Limitations.** Researchers should note that the MagDM method proposed in this paper has some limitations. One such limitation is its dependence on the choice of asymmetric kernel functions, which can impact its performance. Additionally, MagDM may be computationally expensive for large datasets, as it requires $\mathcal{O}(N^2)$ memory and $\mathcal{O}(N^3)$ computational complexity to derive the spectral decomposition. However, this limitation can be addressed through the use of out-of-sample extensions, which are discussed in Appendix E.

## E  Out-of-sample extensions.

Out-of-sample extensions are useful in many applications where low-dimensional embeddings computed on the original dataset are extended to new data. The Nyström extension is a well-known technique used in the machine learning community to approximate the Gram matrix by a low-rank embedding. However, the out-of-sample extension of duffision maps for asymmetric kernels has not been studied before. Here, we present the corresponding Nyström-based extension for out-of-sample cases. As discussed earlier, the integral operator (5) is compact and self-adjoint, whose spectral decomposition is $\{\lambda_n^{(q)}, \phi_n^{(q)}\}$. If $\lambda_n^{(q)} \neq 0$, the following identity holds for $\boldsymbol{x} \in X$:

$$\phi_n^{(q)}(\boldsymbol{x}) = \frac{T^{(q)}}{\lambda_n^{(q)}} \phi_n^{(q)}(\boldsymbol{x}) = \int_X \frac{\rho(\boldsymbol{x}, \boldsymbol{y}, q)}{\lambda_n^{(q)}} \phi_n^{(q)}(\boldsymbol{y}) \mathrm{d}\mu(\boldsymbol{y}) \, .$$

The Nyström extension extends the equation above to new data $Z$ such that $X \subseteq Z$ as follows,

$$\phi_n^{(q)}(\boldsymbol{z}) = \int_X \frac{\rho(\boldsymbol{z}, \boldsymbol{y}, q)}{\lambda_n^{(q)}} \phi_n^{(q)}(\boldsymbol{y}) \mathrm{d}\mu(\boldsymbol{y}) \, , \tag{a1}$$

where $\boldsymbol{z} \in Z$ and $\phi_n^{(q)}(\boldsymbol{z}) = \sum_{\boldsymbol{y} \in X} \frac{\rho(\boldsymbol{z}, \boldsymbol{y}, q)}{N \lambda_n^{(q)}} \phi_n^{(q)}(\boldsymbol{y})$ is the empirical form of (a1) for $X$. This allows the eigenfunctions to be extended for new data, enabling the extension of MagDM (8) as follows,

$$\psi^{t,(q)}(\boldsymbol{z}) = \sum_{\boldsymbol{y} \in X} \left[ \frac{\rho(\boldsymbol{z}, \boldsymbol{y}, q)}{N} \phi_1^{(q)}(\boldsymbol{y}), \quad \frac{\rho(\boldsymbol{z}, \boldsymbol{y}, q)}{N} \phi_2^{(q)}(\boldsymbol{y}), \quad \cdots, \quad \frac{\rho(\boldsymbol{z}, \boldsymbol{y}, q)}{N} \phi_{s(\delta,t)}^{(q)}(\boldsymbol{y}) \right]^\top \, .$$

## F  Descriptions and figures of datasets

In this section, we visualize the datasets using either expert knowledge or a force-directed layout. We hope that these visualizations will help readers gain a better understanding of the data.

### F.1  The first artificial network

Fig. Ape.2(a) provides the running flow of the first artificial network and Fig. Ape.2(b) shows an example of the directed graphs generated with $P = 0$, where the asymmetric adjacency connection can be considered as an asymmetric kernel.

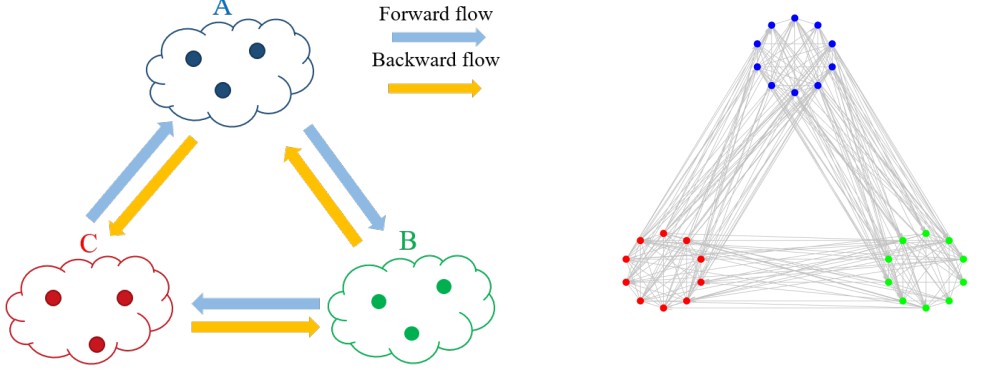

(a) The running flow of three groups.      (b) Graph using the expert knowledge positions.

Figure Ape.2: An illustration of the first artificial network. (a) The running flow of three groups A, B and C. The directed/asymmetric information is nested in the running flow. (b) An instance of a directed graph generated by the running flow with backward flow probability $P = 0$.

## F.2 The second artificial network

The running flow of the second artificial network comprises four groups (A, B, C, and D). The structure of the flow is apparent, with groups A and D serving as out-come and in-come nodes, respectively, while groups B and C function as communicators. Groups B and C are two dense sets containing 20 nodes and Group A and D are a pair of datasets whose interconnections are much more than Group B and C.

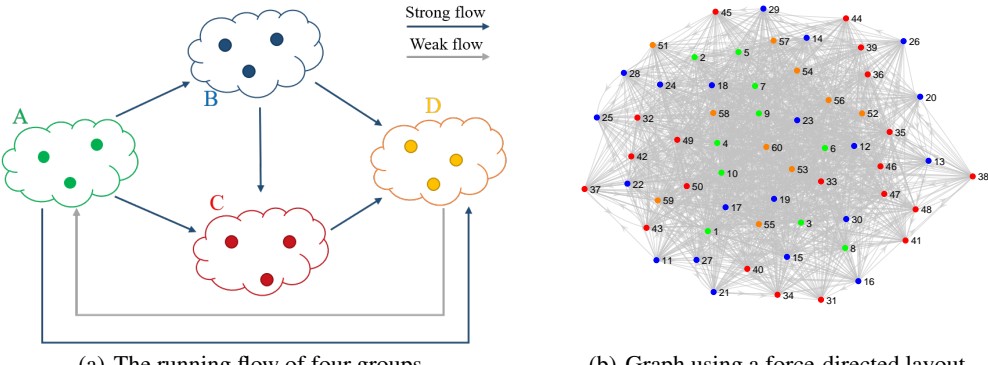

(a) The running flow of four groups.     (b) Graph using a force-directed layout.

Figure Ape.3: An illustration of the second artificial network. (a) The running flow of three groups A, B, C and D. The directed/asymmetric information is nested in the running flow. (b) An instance of a directed graph generated by the running flow with Groups A and D playing a particular role (green and orange nodes) and Groups B and C playing a role of a communicator (red and blue nodes).

## F.3 The Möbius strip

The Möbius strip dataset is a set of 300 points randomly distributed along the Möbius strip. The parametric form of the Möbius strip is defined by,

$$x(u,v) = \left(1 + \frac{v}{2}\cos\frac{u}{2}\right)\cos u, \quad y(u,v) = \left(1 + \frac{v}{2}\cos\frac{u}{2}\right)\sin u, \quad z(u,v) = \frac{v}{2}\sin\frac{u}{2},$$

where $0 \le u \le 2\pi$ and $-0.5 \le v \le 0.5$. The dataset is with a color drift in the counterclockwise direction on the x-y plane in Fig. Ape.4.

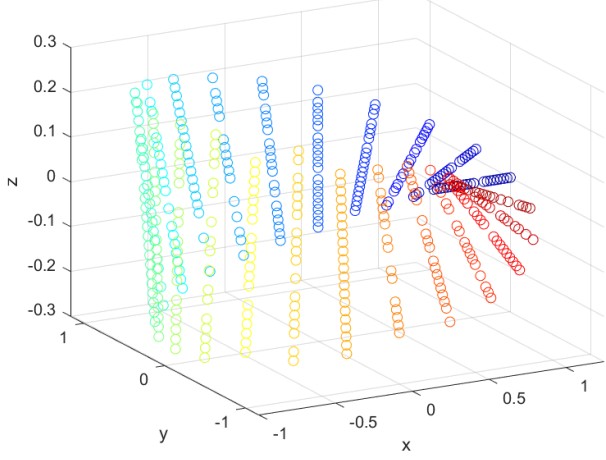

Figure Ape.4: Dataset with 300 random points in the Möbius strip.

## F.4    Two trophic networks

We have chosen two specific trophic networks: the Mondego [32] and Florida [33] networks, which are part of the Pajek datasets. These networks have recorded the trophic exchanges at Mondego estuary and Florida bay during the wet season, respectively. Based on the roles of the nodes in these ecosystems, we have classified them into different categories, as shown in Appendix Ape.5. The green nodes, such as 2um Spherical Phytoplankt and Phytoplankton, are producers that generate their own food through photosynthesis or chemosynthesis. The brown and red nodes are low-level consumers like littorina and high-level consumers like bonefish and crocodiles that feed on other organisms for energy. Additionally, the purple node in the Florida network represents decomposers that break down dead or decaying organic matter. Finally, the blue and turquoise blue nodes correspond to the input, output, and organic matter of the ecosystem, respectively.

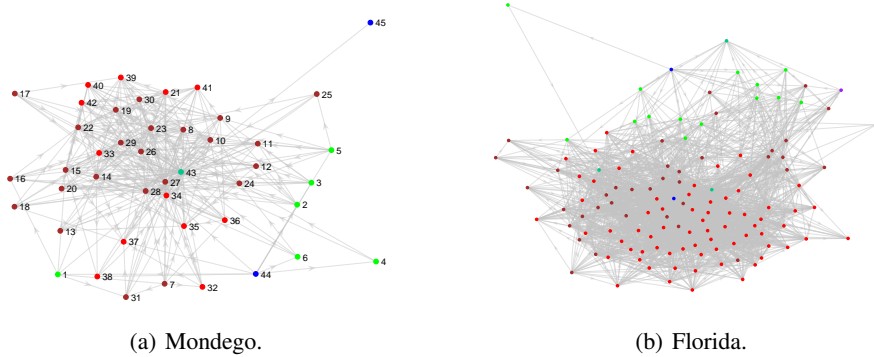

(a) Mondego.

(b) Florida.

Figure Ape.5: The trophic networks using force-direct layout. Nodes among this network are classified into several categories, The green nodes are producers that generate their own food through photosynthesis or chemosynthesis. The brown and red nodes are low-level consumers and high-level consumers that feed on other organisms for energy. Additionally, the purple node in the Florida network represents decomposers that break down dead or decaying organic matter. The blue and turquoise blue nodes correspond to the input, output, and organic matter of the ecosystem.

# G  Quantitative experiments

We have incorporated quantitative experiments to measure the performance of the proposed method (MagDM) in comparison to two other methods (ME and MME). In the first network, we set the forward flow probability to 0.5 and the backward flow probability to P=0/0.2/1. We then cluster the low-dimensional embeddings of the three methods using the k-means algorithm with k=3. To evaluate performance for the clustering results, we consider two internal evaluation metrics: the silhouette coefficient (SC) and the Davies-Bouldin index (DB), as well as two external evaluation metrics: the adjusted Rand index (ARI) and the normalized mutual information (NMI). These metrics assess the quality of clustering.

The results of the quantitative experiments evaluating the performance of the proposed MagDM are presented in Tab. Ape.1 of the attached PDF file. It can be seen that MagDM achieves higher scores in the metrics. Particularly when P=1, where interconnections are more complicated, MagDM significantly outperforms other methods, which highlights the superior effectiveness and robustness of MagDM compared to other methods.

Table Ape.1: Quantitative experiments on the first network in Section 5.1. The symbol ↑/↓ indicates that higher/lower scores correspond to better clustering results.

| Evaluation metrics | | MagDM | ME | MME |
|---|---|---|---|---|
| P=0 | SC↑ | 0.8583±0.0461 | 0.8444±0.0251 | **0.8616±0.0694** |
| | DB↓ | 0.3647±0.0610 | 0.3918±0.0709 | **0.3525±0.0587** |
| | ARI↑ | **0.9365±0.0581** | 0.9085±0.1051 | 0.9285±0.0586 |
| | NMI↑ | **0.9617±0.0624** | 0.9227±0.0529 | 0.9548±0.0481 |
| P=0.2 | SC↑ | **0.8839±0.0408** | 0.8701±0.0168 | 0.8788±0.0447 |
| | DB↓ | 0.3277±0.0717 | 0.3343±0.0625 | **0.3012±0.0805** |
| | ARI↑ | **0.8940±0.0718** | 0.8863±0.0756 | 0.86770.0626 |
| | NMI↑ | **0.9106±0.1115** | 0.8608±0.0762 | 0.8259±0.0871 |
| P=1 | SC↑ | **0.8206±0.0423** | 0.7987±0.0342 | 0.7979±0.0354 |
| | DB↓ | **0.3924±0.0396** | 0.4311±0.0848 | 0.4269±0.0777 |
| | ARI↑ | **0.9266±0.0888** | 0.2555±0.1818 | -0.0030±0.0091 |
| | NMI↑ | **0.9484±0.0667** | 0.2124±0.1716 | 0.0074±0.0061 |

# H  Experiments regarding the importance of the parameter $t$

We have conducted additional experiments on MagDM, focusing on diffusion time $t \in \{1, 2, 3, 4, 5, 10\}$. The results are presented in Fig. Ape.6, which can be found in the attached PDF file under the General Response section.

In Fig. Ape.6, we illustrate the clusters that evolve from different values of $t$ ($t \in \{1, 2, 3, 4, 5, 10\}$). At $t = 1$, the local geometry becomes apparent with four clusters visible on the real and imaginary axes. Due to the presence of asymmetric connections, the diffusion distance between samples within the same group is small, while the diffusion distance between samples in different groups is large. Consequently, even with reduced dimensions, we can clearly distinguish the four clusters in the diffusion space. As the diffusion process progresses, the diffusion distance between different groups decreases. At $t = 4$, the groups start to connect with each other, forming a single structure. Finally, at $t = 10$, the four groups merge to create a single super-cluster structure, with very small diffusion distances between points. Interestingly, the clustering on the phase axes remains clear and preserved throughout the diffusion time. This observation demonstrates the strong ability of MagDM to capture asymmetric information during the diffusion process. The dynamic behavior we observe underscores the effectiveness of representing and visualizing the data.

# I  Discussion of the diffusion framework

The MagDM framework, proposed for asymmetric scenarios, employs a mathematical technique to transform a pair of real-valued asymmetric similarities into complex-valued and conjugate symmetric similarities. By leveraging the concept of random walks on graphs, the framework computes the diffusion distance $D^t(x, y)$. This distance is a functional weighted $L^2$ distance between complex-valued proximities, enabling a more flexible and adaptive measure of similarity. $D^t(x, y)$ will be

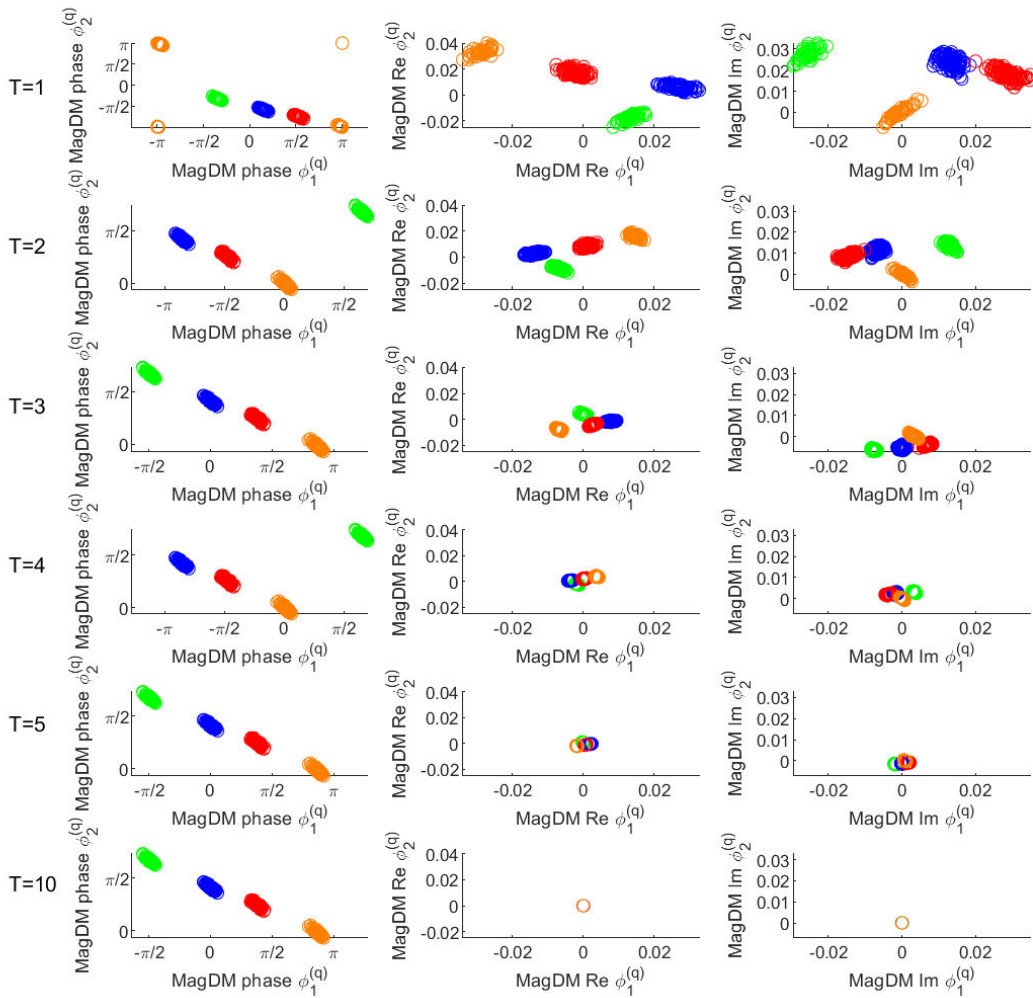

Figure Ape.6: Clusters evolving from $t \in \{1, 2, 3, 4, 5, 10\}$ on the four groups colored by red, green, blue, and orange, with $q = 1/3$. For better illustration, the real and imaginary parts of the first two non-trivial eigenvectors are reported, in addition to the phases.

small if two samples are similar in the complex-valued plane and vice versa. Consequently, it becomes capable of capturing both local and global patterns within the data.

In this framework, the diffusion kernel, denoted as $\rho(x, y)$, represents the transition probabilities of the random walks. It signifies the likelihood of transitioning from one data point to another within the network. In summary, the proposed diffusion distance within the MagDM framework facilitates a geometric interpretation of asymmetric similarity. It encompasses the network structure and takes into account the asymmetric relationships between data points.

## J   The comparison with [4]

The MagDM is not a simple derivation of diffusion map for $U(1)$-connection graphs because there are three key differences between MagDM and the graph connection Laplacian (GCL) approach.

**Definition of the similarity matrices.** MagDM focuses on asymmetric similarity between samples and calculates the $U(1)$ transporter for all pairs of samples. On the other hand, GCL focuses on symmetric cases and calculates the orthogonal $O(d)$ transporter for graph edges. While there is an isomorphism between $U(1)$ and $SO(2)$, they differ in implementation. MagDM measures scalar

similarity between two samples, denoted as x and y, while the similarity between x and y in GCL is a $d * d$ block. The corresponding Gram matrix in MagDM, denoted by $\rho$, has a size of $N * N$, while the corresponding Gram matrix in GCL, denoted by $S$, is $Nd * Nd$.

**Definition of diffusion distance.** In MagDM, the diffusion distance is defined as $[D^t(x, y)]^2 = \int_X (\rho^t(x, u) - \rho^t(y, u))^2 \mu(du)$. It measures the number of paths of length t connecting x and y in the complex-valued plane. The diffusion distance is calculated by the sum of squares of the difference between $\rho^t(x, \bullet)$ and $\rho^t(y, \bullet)$. In GCL, diffusion distance has a different form: $[D^t(x, y)]^2 = Tr(S^t(x, y)S^t(x, y)^\top)$, which measures the agreement between transporters and the number of paths of length t connecting x and y. Based on these two definitions, it can be observed that $U(1)$-connection graphs exhibit different diffusion distances in MagDM and GCL.

**Definition of the diffusion map.** Due to the differences in diffusion distances, the diffusion maps of the two methods also differ. In MagDM, eigenvalues and eigenvectors are defined as $\{\lambda_i\}_{i=1}^N, \{\phi_i\}_{i=1}^N$, respectively. The diffusion map is defined as $\Phi^t(x) = (\lambda_1^t \phi_1(x), \lambda_2^t \phi_2(x), \cdots, \lambda_N^t \phi_N(x))$. For vector diffusion map, its eigenvalues and eigenvectors are defined as $\{\eta_i\}_{i=1}^{Nd}, \{\psi_i\}_{i=1}^{Nd}$, respectively. Its diffusion map is defined by $\Phi^t(x) = ((\eta_i \eta_j)^t < \psi_i(x), \psi(x) >)_{i,j=1}^{Nd}$. These two diffusion maps have different forms from each other. Overall, MagDM is not a derivation to a notion of diffusion map for $U(1)$-connection graphs.

# K    Eigenvalues and robustness

Indeed, $H$ and $I - H$ can be transferred to each other when considering both positive and negative eigenvalues. However, it should be noted that in [18, 19], the negative eigenvalues are truncated, which means that $H$ and $I - H$ are not identical in this regard. We focus on the k eigenvalues that include both positive and negative eigenvalues with the largest absolute magnitudes. These top k eigenvalues correspond to the slowest diffusion processes or the global structures in the data. It is also believed that the larger magnitude of eigenvalues corresponds to better discriminative information, as discussed in [37]. One notable advantage of this approach is its robustness when confronted with perturbed asymmetric similarity. Even if the similarity measure is distorted, we can still obtain the principal components using this method. However, there is no specific reference to a related study because most previous work has mainly focused on positive semi-definite matrices. When dealing with asymmetry, it becomes necessary to consider negative eigenvalues. This aspect is worth exploring in future research. Currently, we can observe this phenomenon experimentally in Fig. 1 of the manuscript. In the case where P=0, it represents a situation where there are only links in one direction, making it an asymmetric but relatively easy scenario. When P=1/0.8, the situation becomes more complicated and backward flow is perturbed by the forward, MagDM effectively distinguishes between the three groups while ME/MME struggles to capture the information, highlighting stronger robustness and effectiveness of MagDM.

In the experiments involving asymmetric similarity, it is indeed possible for the largest eigenvalues to have different signs. Here, we present the counts of eigenvalues for $\phi_1$ and $\phi_2$ that exhibit both positive signs (++) and opposite signs (+-) among 100 trials for the network generated by the forward and backward flows (Experiment 1). The forward flow probability is fixed at 0.5, while the backward flow probability P is selected from the following set: $\{0, 0.3, 0.4, 0.5, 0.6, 0.65, 0.675, 0.7, 0.8, 1\}$.

Table Ape.2: The counts of signs for different P.

| P   | 0   | 0.3 | 0.4 | 0.5 | 0.6 | 0.65 | 0.675 | 0.7 | 0.8 | 1   |
| --- | --- | --- | --- | --- | --- | ---- | ----- | --- | --- | --- |
| ++  | 100 | 100 | 96  | 90  | 81  | 57   | 16    | 2   | 0   | 0   |
| +-  | 0   | 0   | 4   | 10  | 19  | 43   | 84    | 98  | 100 | 100 |

In Tab. Ape.2, it can be observed that as P increases, the number of opposite signs also increases. Particularly, when P is larger than 0.675, opposite signs occur more frequently. We have discovered that the principal components align with both positive and negative eigenvalues, highlighting the superior performance of MagDM. These results further substantiate the robustness of MagDM.

