# OpenReview forum: "Diffusion Representation for Asymmetric Kernels via Magnetic Transform"
_NeurIPS.cc/2023/Conference — NeurIPS 2023 poster_

### Official Review · Reviewer_E65Y · 2023-07-03

**Soundness:** 4 excellent
**Presentation:** 4 excellent
**Contribution:** 3 good
**Rating:** 6
**Confidence:** 5

**Summary:**

The paper introduces the concept of the magnetic transform to define diffusion representation and diffusion distance for asymmetric kernels. The key idea is to transform an asymmetric kernel into a Hermitian one, enabling the application of standard techniques such as eigen-decomposition.

By leveraging this magnetic transform approach, the authors conduct experimental validations to assess the effectiveness of their proposed method. The results demonstrate that their method outperforms other dimension reduction methods in terms of separating clusters in asymmetric data.

**Strengths:**

- The presentation of the paper is clear and the paper is easy to follow.
- A proper discussion on the selection of the scaling parameter in included which is helpful for people to implement the method.
- The experimental results support the effectiveness of the proposed method.

**Weaknesses:**

A major concern regarding the paper relates to its novelty and theoretical justification. Several points raise doubts regarding the originality and uniqueness of the proposed approach:
  - The Magnetic transform, as presented in the paper, may not be considered entirely novel. Previous works, specifically [18] and [19], have already studied similar forms of the Magnetic transform, particularly when the kernel represents the adjacency matrix of a directed graph. This raises questions about the extent to which the proposed approach differs from previous research. While the paper focuses on a different matrix form ($H$ instead of $D-H$ or $I-H$), more theoretical justification is needed to demonstrate the significance and appropriateness of this choice.
  - Besides, I would like to bring the authors' attention to the following paper: ```VECTOR DIFFUSION MAPS AND THE CONNECTION LAPLACIAN``` by Singer and Wu. It is well known that magnetic laplacian is the same as the connection laplacian when considering $SO(2)$ signatures. In this reference, Singer and Wu have already considered diffusion maps and diffusion distances for connection graphs which could well be a generalization already for what the authors are proposing. I think a comparison with this reference is needed.

Additionally, there is a lack of clarity in the experiments regarding the importance of the parameter $t$ and the role of diffusion in the proposed method. The paper does not explicitly demonstrate how the choice of $t$ influences the results or why diffusion is significant. In fact, the current experiments only show that the eigenvectors of the normalized kernel is useful. This ambiguity hampers a thorough understanding of the method and its underlying principles.

**Questions:**

- line 86: why do you choose $t$ to be an integer instead of a real number?
- What is the kernel involved in the experiments in Section 5.1?

---

> ### Author Rebuttal · Authors · 2023-08-09
>
> Thanks for your careful reviewing and insightful suggestions. We address your concerns as below:
>
> **R3.1 Contributions of (Eq.3).**
>
> Thank you for your insightful thoughts on the difference from existing works. The contribution of our work lies in the development of a diffusion representation framework designed for asymmetric kernels. In contrast, existing works [1,2] primarily focus on asymmetric matrices. It is crucial to note that these two approaches are fundamentally distinct: our framework naturally produces a diffusion representation and function decomposition customized to a matrix derived from a kernel function, whereas the reverse may not hold true. This development was facilitated by exploring integral operators of magnetic transform kernels, and Proposition 1 provides the condition for the existence of spectral decomposition.
>
> The magnetic transform is designed for asymmetric kernel cases. It is built upon the observation made in our manuscript, supported by Proposition 2, that the eigenvalues of the kernel function H are not uniformly positive, serving as the basis for our design of MagDM. Based on the observation, it is apparent that asymmetric information is embedded not only in positive eigenvalues but also in negative eigenvalues, along with their corresponding eigenfunctions. By incorporating both positive and negative eigenvalues of H, we achieve enhanced robustness and effectiveness. In contrast, [1,2] (I-H) only focused on positive eigenvalues, neglecting the importance of this phenomenon. As a consequence, their approach suffered from limited robustness. The experimental results support the effectiveness of MagDM.
>
> **R3.2 The comparison with Singer and Wu's work [3].**
>
> We appreciate your insightful comment regarding Singer and Wu's work [3] which indeed provides a valuable foundation in the field. However, it is important to clarify that MagDM and [3] are not generalizations of each other.
>
> The proposed methods have three fundamental differences with [3]. Firstly, the addressed problems of them are different. The proposed method addresses the challenge of dealing with asymmetric kernels while [3] focuses on symmetric kernels. Secondly, the unitary transporter defined in (Eq. 3) is utilized for all pairs of samples which reflects asymmetric geometries of the data. In contrast, the SO(2) signatures in [3] are optimized only for nearby samples as best rotational alignments. [3] suggests that these samples, denoted as $x$ and $y$, should satisfy the condition $0 < \\| x - y \\|_{\mathbb{R}^p} < \sqrt{\epsilon}$, where $\epsilon$ is the scaling parameter. And the connection Laplacian is the convergence in the limit of $N \rightarrow \infty$ and $\epsilon \rightarrow 0$ of the Vector Diffusion Map (VDM). In this scenario, SO(2) signatures are optimized only for pairs of samples that are very close to each other in Euclidean space. Thirdly, We employ a complex and unitary transporter in U(1), which is controlled by q. In [3], the SO(2) transporter is determined through local PCA. Additionally, the diffusion distance (Eq.6) in MagDM represents a functional weighted distance between complex-valued proximity of two samples, and the corresponding diffusion map is also complex-valued. In contrast, [3] operates in real-valued working spaces. A similar discussion is also reported in Section 3.3 of the paper [1] which explains the connection between ME and VDM.
>
> These distinctions limit the applicability of [3] to the problem addressed in this paper. However, MagDM offers a viable approach to handling asymmetric kernels. We appreciate you highlighting this point, and we will ensure to include these clarifications in our research.
>
> **R3.3 The lack of clarity of the parameter in the experiments.**
>
> Thank you for your insightful comments. We have conducted additional experiments on MagDM, focusing on diffusion time $t$. The results are presented in Fig. R1 of the attached PDF file under Author Rebuttal.
>
> In Fig. R1, we illustrate the clusters that evolve from different values of $t\in\\{1,2,3,4,5,10\\}$. At $t=1$, the local geometry becomes apparent with four clusters visible on the real and imaginary axes. The diffusion distance between samples within the same group is small, while the diffusion distance between samples in different groups is large. As the diffusion process progresses, the diffusion distance between different groups decreases. At $t=4$, the groups start to connect with each other, forming a single structure. Finally, at $t=10$, the four groups merge to create a single super-cluster structure, with very small diffusion distances between points. Interestingly, the clustering on phase axes remains clear and preserved throughout $t$. This observation shows the strong ability of MagDM to capture asymmetric information during the diffusion process. The dynamic behavior we observe underscores the effectiveness of representing and visualizing the data. We will ensure to include the experiments in Appendix of the manuscript due to the space limitation.
>
> **R3.4 Discussion about diffusion time $t$.**
>
> Thanks for your careful reading. We acknowledge that there is a typo and $t$ is chosen to be a positive real number. We assure you that it will be rectified in the revised version.
>
> **R3.5 Kernels in Section 5.1**
>
> We utilize the adjacency matrices as Gram matrices of the data. The adjacency matrices are generated by the running flow probability. $K_{ij}=1$ if sample $x_i$ is to connect sample $x_j$ and $K_{ij}=0$ if not.
>
> **Referneces**
>
> [1] Fanuel M, et al. Magnetic eigenmaps for the visualization of directed networks[J]. ACHA, 2018, 44(1): 189-199.
>
> [2] Cloninger A. A note on markov normalized magnetic eigenmaps[J]. ACHA, 2017, 43(2): 370-380.
>
> [3] Singer A, Wu H T. Vector diffusion maps and the connection Laplacian[J]. Commun Pure Appl Math, 2012, 65(8): 1067-1144.

---

> > ### Comment · Reviewer_E65Y · 2023-08-12
> >
> > I want to extend my thanks to the authors for addressing my questions. However, I would like to further clarify one of my concerns and pose another question for the authors to elucidate.
> >
> > 1. **On Eigenvalues and Robustness**: Could the authors provide more insights into why incorporating both negative and positive eigenvalues can enhance robustness? Mathematically speaking, the eigenvalues of $H$ and $I-H$ are identical, except for a change of sign and a constant shift. Is there any reference to a related study that can shed light on this aspect? Additionally, the experiments presented don't seem to align with the authors' claim of robustness. The focus on the first two eigenvectors leaves me uncertain about how both positive and negative eigenvalues were utilized. More details on this would be greatly appreciated.
> >
> > 2. **On Connection Laplacian**: I would like to specify my concern regarding the connection Laplacian. I was not referring to the connection Laplacian in the context of Riemannian geometry, but rather to the concept of graph connection Laplacian, where groups are associated with graph edges (see [1,2] for terminology). In Section 3 of [3], the authors introduce a diffusion map defined for general $O(d)$-connection graphs (this is not restricted to the connection graphs generated by local PCA). Since there is an isomorphism between $U(1)$ and $SO(2)$, there appears to be an obvious derivation to a notion of diffusion map for $U(1)$-connection graphs. This is the aspect I would like the authors to address.
> >
> >
> > [1] Fanuel and Bardenet, 2022. Sparsification of the regularized magnetic Laplacian with multi-type spanning forests
> >
> > [2] Bandeira et at. 2013. A Cheeger Inequality for the Graph Connection Laplacian
> >
> > [3] Singer and Wu, 2011. Vector diffusion maps and the connection Laplacian

---

> > > ### Author Response · Authors · 2023-08-13
> > >
> > > I would like to express my sincere gratitude to the reviewer for the valuable feedback and insightful comments.
> > >
> > > **R3.1 Eigenvalues and Robustness**
> > >
> > > Thanks for your insightful comment. Indeed, $H$ and $I-H$ can be transferred to each other when considering both positive and negative eigenvalues. However, it should be noted that in [1,2], the negative eigenvalues are truncated, which means that $H$ and $I-H$ are not identical in this regard. We focus on the k eigenvalues that include both positive and negative eigenvalues with the largest absolute magnitudes. These top k eigenvalues correspond to the slowest diffusion processes or the global structures in the data. It is also believed that the larger magnitude of eigenvalues corresponds to better discriminative information, as discussed in [3]. One notable advantage of this approach is its robustness when confronted with perturbed asymmetric similarity. Even if the similarity measure is distorted, we can still obtain the principal components using this method. However, there is no specific reference to a related study because most previous work has mainly focused on positive semi-definite matrices. When dealing with asymmetry, it becomes necessary to consider negative eigenvalues. This aspect is worth exploring in future research. Currently, we can observe this phenomenon experimentally in Fig. 1 of the manuscript. In the case where P=0, it represents a situation where there are only links in one direction, making it an asymmetric but relatively easy scenario. When P=1/0.8, the situation becomes more complicated, and backward flow is perturbed by the forward, MagDM effectively distinguishes between the three groups while ME/MME struggles to capture the information, highlighting the stronger robustness and effectiveness of MagDM.
> > >
> > > [1] Fanuel M, et al. Magnetic eigenmaps for the visualization of directed networks[J]. ACHA, 2018, 44(1): 189-199.
> > >
> > > [2] Cloninger A. A note on Markov normalized magnetic eigenmaps[J]. ACHA, 2017, 43(2): 370-380.
> > >
> > > [3] Hamm J, Lee D D. Grassmann discriminant analysis: a unifying view on subspace-based learning[C]. ICML. 2008: 376-383.
> > >
> > > **R3.2 Discussion about Connection Laplacian**
> > >
> > > Thank you for kindly specify the concern. The MagDM is not a simple derivation of diffusion map for $U(1)$-connection graphs because there are three key differences between MagDM and the graph connection Laplacian (GCL) approach.
> > >
> > > 1. Definition of the similarity matrices.
> > >
> > >     MagDM focuses on asymmetric similarity between samples and calculates the $U(1)$ transporter for all pairs of samples. On the other hand, GCL focuses on symmetric cases and calculates the orthogonal $O(d)$ transporter for graph edges. While there is an isomorphism between $U(1)$ and $SO(2)$, they differ in implementation. MagDM measures scalar similarity between two samples, denoted as x and y, while the similarity between x and y in GCL is a $d\*d$ block. The corresponding Gram matrix in MagDM, denoted by $\rho$, has a size of $N\*N$, while the corresponding Gram matrix in GCL, denoted by $S$, is $Nd \* Nd$.
> > >
> > > 2. Definition of diffusion distance.
> > >
> > >    In MagDM, the diffusion distance is defined as $[D^t(x,y)]^2=\int_X(\rho^t(x,u)-\rho^t(y,u))^2\mu(du)$. It measures the number of paths of length t connecting x and y in the complex-valued plane. The diffusion distance is calculated by the sum of squares of the difference between $\rho^t\left(x,\bullet\right)$ and $\rho^t\left(y,\bullet\right)$. In GCL, diffusion distance has a different form: $[D^t(x,y)]^2=Tr(S^t(x,y)S^t(x,y)^\top)$, which measures the agreement between transporters and the number of paths of length t connecting x and y. Based on these two definitions, it can be observed that $U(1)$-connection graphs exhibit different diffusion distances in MagDM and GCL.
> > >
> > > 3. Definition of the diffusion map.
> > >
> > >       Due to the difference in diffusion distances, the diffusion maps of the two methods also differ. In MagDM, eigenvalues and eigenvectors are defined as $\\{\\lambda_i\\}\_{i=1}^N,\\{\\phi_i\\}\_{i=1}^N$, respectively. The diffusion map is defined as $\Phi^t(x)=(\lambda_1^t\phi_1\left(x\right),\lambda_2^t\phi_2\left(x\right),\cdots,\lambda_N^t\phi_N(x))$. For vector diffusion map, its eigenvalues and eigenvectors are defined as $\\{η_i\\}\_{i=1}^{Nd},\\{ψ_i\\}\_{i=1}^{Nd}$, respectively. Its diffusion map is defined by $\Phi^t(x)=((η_iη_j)^t < ψ_i(x),ψ_j(x) > )_{i,j=1}^{Nd}$. These two diffusion maps have different forms from each other. Overall, MagDM is not a derivation to a notion of diffusion map for $U(1)$-connection graphs.

---

> > > > ### Comment · Reviewer_E65Y · 2023-08-16
> > > >
> > > > Thank you for the elaborative responses and for addressing the comparison with the connection Laplacian. This certainly enhances the paper's depth, and I've adjusted my score accordingly.
> > > >
> > > > Regarding the eigenvalues discussion in the reply, I'd like to seek further clarification:
> > > >
> > > > I may have missed it in the main text, but from your experiments, am I right in understanding that  $\phi_1$ and $\phi_2$ are selected based on the two eigenvalues with the largest absolute magnitudes?
> > > >
> > > > To better appreciate your stance on the advantage of utilizing both positive and negative eigenvalues, could you confirm if, in your experiments, $\phi_1$ and $\phi_2$ typically correspond to eigenvalues of opposite signs?

---

> > > > > ### Author Response · Authors · 2023-08-16
> > > > >
> > > > > We deeply appreciate your efforts in re-evaluating the novelty of our work.
> > > > >
> > > > > 1. Yes, in MagDM, eigenvectors are selected based on the eigenvalues with the largest absolute magnitudes.
> > > > >
> > > > > 2. Your guess is correct. In the experiments involving asymmetric similarity, it is indeed possible for the largest eigenvalues to have different signs. Here, we present the counts of eigenvalues for $\phi_1$ and $\phi_2$ that exhibit both positive signs (++) and opposite signs (+-) among 100 trials for the network generated by the forward and backward flows (Experiment 1). The forward flow probability is fixed at 0.5, while the backward flow probability P is selected from the following set: {0, 0.3, 0.4, 0.5, 0.6, 0.65, 0.675, 0.7, 0.8, 1}.
> > > > >
> > > > >    Table A: The counts of signs for different P.
> > > > >     P  | 0   | 0.3 | 0.4 | 0.5 | 0.6 | 0.65 | 0.675 | 0.7 | 0.8 | 1
> > > > >     :----:|:-----:|:-----:|:-----:|:-----:|:-----:|:------:|:-------:|:-----:|:-----:|:-----:
> > > > >     ++ | 100 | 100 | 96  | 90  | 81  | 57   | 16    | 2   | 0   | 0
> > > > >     +- | 0   | 0   | 4   | 10  | 19  | 43   | 84    | 98  | 100 | 100
> > > > >
> > > > >     In Table A, it can be observed that as P increases, the number of opposite signs also increases. Particularly, when P is larger than 0.675, opposite signs occur more frequently. This finding also addresses Reviewer SsqM's query regarding the performance of MagDM. We have discovered that the principal components align with both positive and negative eigenvalues, highlighting the superior performance of MagDM. These results further substantiate the robustness of MagDM. We sincerely appreciate the valuable discussion with you.

---

> > > > > > ### Comment · Reviewer_E65Y · 2023-08-19
> > > > > >
> > > > > > Thank you for the detailed clarification. I find the discussed phenomenon interesting. Incorporating these discussions into the paper will undoubtedly enhance its value. I have decided to increase my score one more time for the paper.

---

> > > > > > > ### Author Response · Authors · 2023-08-19
> > > > > > >
> > > > > > > The discussions are indeed helpful and insightful. Your expertise and input have been instrumental in improving our manuscript. We will certainly include them in the final version. We are very grateful for your time and effort in reviewing our manuscript.

---

### Official Review · Reviewer_sG91 · 2023-07-07

**Soundness:** 3 good
**Presentation:** 2 fair
**Contribution:** 3 good
**Rating:** 7
**Confidence:** 2

**Summary:**

This paper studies the asymmetric kernel case for the diffusion map. The authors utilize the magnetic transform technique to develop a diffusion representation framework for the asymmetric kernel case. They investigate several properties of the proposed magnetic transform kernel. The challenge lies in defining the corresponding integral operators and diffusion geometry. In their experiments, the authors show their proposed MagDM framework competitive and even superior to existing dimension reduction techniques like DM, KPCA, etc.

**Strengths:**

This work combine the magnetic transform and the diffusion map techniques to handle the asymmetric kernel case of the diffusion map. It seems to be a pioneer research work by defining new concepts.

**Weaknesses:**

All the experiments present only qualitative results and some quantitative results would help figure out the advantages of the proposed MagDM framework over other existing techniques.

**Questions:**

I am not familiar with the topic and feel quite confused about Proposition 1: why do we want to assume a Hilbert-Schmidt kernel $X$ and define another Hilbert-Schmidt kernel $H^{(q)}$? The goal seems to define a kernel with conjugate symmetry.

**Limitations:**

The authors have mentioned some limitations of the proposed MagDM framework in appendix. For example, different choices of asymmetric kernel functions would impact the performance of the framework. Currently, there is no clue about how to handle it.

---

> ### Author Rebuttal · Authors · 2023-08-09
>
> Thanks for appreciating the novelty of our work and for providing insightful comments. We address your concerns as below:
>
> **R2.1 Some quantitative results would help figure out the advantages of the proposed method.**
>
> We sincerely appreciate your insightful comments. In response, we have conducted the quantitative experiments as suggested. For further details, please refer to **R0.1** in the Author Rebuttal.
>
> **R2.2 The goal of defining another Hilbert-Schmidt kernel.**
>
> Defining the Hilbert-Schmidt kernel is a preparatory step for MagDM. The goal is to define the Hermitian kernel function whose spectral decomposition is utilized to support the proposed diffusion maps (Eq.8).  The existence of the spectral decomposition requires the corresponding kernels to be both conjugate symmetric and Hilbert-Schmidt kernels. Therefore, it is necessary to define the Hilbert-Schmidt kernel $H^{(q)}$ based on Proposition 1.

---

> > ### Comment · Reviewer_sG91 · 2023-08-19
> >
> > Thank you for your clarification and I will keep my score as 7.

---

> > > ### Author Response · Authors · 2023-08-19
> > >
> > > We deeply appreciate your valuable feedback and insightful comments. We would keep improving our manuscript to involve your insights.

---

### Official Review · Reviewer_SsqM · 2023-07-08

**Soundness:** 3 good
**Presentation:** 3 good
**Contribution:** 4 excellent
**Rating:** 7
**Confidence:** 4

**Summary:**

The paper proposes proposes a new method called MagDM in which it connects Diffusion maps (DM) and the Magnetic transform (MT). DM is a nonlinear dimension reduction technique obtains a lower dimensional embedding by using the information of the diffusion distances that assume symmetry. However, in practice the intrinsic and extrinsic geometries of data can be asymmetric. MT is a promising technique that converts an asymmetric matrix to a Hermitian one, make it suitable for working with asymmetry, but the connection between DM and MT haven’t been explored much. Hence, the main contribution of the paper is to make such connections between DM and MT. Specifically, they developed a diffusion framework endowed with asymmetric kernels using MT. Also, the integral operators of MT, whose compactness is established if the asymmetric kernels are Hilbert-Schmidt kernels have been explored. They’ve also proven an important property that the spectral radius of the proposed integral operator is 1, which ensures that MagDM will not diverge during the diffusion process.

**Strengths:**

The ideas presented here are original. The paper is generally well-written. There are numerous qualitative experiments. Since asymmetric data is common, the proposed approach could have a high significance and impact and could be useful in other methods that use the diffusion framework.

**Weaknesses:**

The paper is missing quantitative experiments. While the qualitative experiments in the paper do suggest that MagDM is capturing the asymmetric geometry better than other approaches, the scatter plots are difficult to interpret without expert knowledge of the underlying data. It would be better to have some numerical evaluation of this. I recommend the authors come up with some metric that measures this performance for comparing the different methods.

Other comments
The diffusion framework could be clarified further. For example, what is the geometric interpretation of the diffusion distances and how do they relate to $\rho(x,y)$?

**Questions:**

1. Can the authors use quantitative measures of how well the methods perform?

2. In Figure 1, it is shown that MagDM has better results compared to ME & MME when P has large values (greater than 0.5). On the other hand, when P is less than 0.5, specifically when P = 0, P = 0.2, all three methods are doing well to separate the groups. So if P = 0 and P = 0.2 are also considered to have high level of asymmetry information, how does MagDM outperform others?

3. In Figure 3, from the perspective of dimension reduction, I would like to compare the result to the original structure . Since the original structure is connected for a M¨obius strip, I would expect the lower dimensional embedding to be connected in a loop as well. However you’ve mentioned in line 210 that you focus on the asymmetric geometry of the data but I am not sure how does the ”asymmetric structure” look like in the original space colored by the rainbow map. So I think you should add the additional subplots to your figures to show us how the original data looks like.

**Limitations:**

Addressed

---

> ### Author Rebuttal · Authors · 2023-08-09
>
> Thanks for your insightful comments and appreciation of the novelty. We address your concerns as below:
>
> **R1.1 Quantitative experiments are needed.**
>
> We sincerely appreciate your insightful comments. In response, we have conducted the quantitative experiments as suggested. For further details, please refer to **R0.1** in the Author Rebuttal.
>
> **R1.2 Discussions of the diffusion framework.**
>
> The MagDM framework, proposed for asymmetric scenarios, employs a mathematical technique to transform a pair of real-valued asymmetric similarities into complex-valued and conjugate symmetric similarities. By leveraging the concept of random walks on graphs, the framework computes the diffusion distance $D^t(x,y)$. This distance is a functional weighted $L^2$ distance between complex-valued proximities, enabling a more flexible and adaptive measure of similarity. $D^t(x,y)$ will be small if two samples are similar in the complex-valued plane and vice versa.
> Consequently, it becomes capable of capturing both local and global patterns within the data.
>
> In this framework, the diffusion kernel, denoted as $\rho(x,y)$, represents the transition probabilities of the random walks. It signifies the likelihood of transitioning from one data point to another within the network. In summary, the proposed diffusion distance within the MagDM framework facilitates a geometric interpretation of asymmetric similarity. It encompasses the network structure and takes into account the asymmetric relationships between data points. We will ensure to include these clarifications in our research.
>
> **R1.3 How does MagDM outperform others in Figure 1?**
>
> Thank you for your insightful observations. Both ME and MME are designed to extract asymmetric geometric information. When P=0, it represents a situation where there are only links in one direction, making it an asymmetric but relatively easy scenario. When P=1/0.8, the situation becomes more complicated, and ME/MME may struggle to capture the information. In contrast, MagDM effectively distinguishes between the three groups in both scenarios, highlighting its stronger robustness and effectiveness. The reason could be intuitively explained as follows: ME/MME focuses solely on the positive part of the eigenvalues of the Hermitian kernel function (Eq.3) but as Proposition 2 demonstrates that the eigenvalues are not necessarily all positive. In the proposed method, regardless of their sign, we take into account all the eigenvalues and the advantages are more obvious for hard tasks, i.e., P=1/0.8.
>
> **R1.4 Results about the M\"obius strip**
>
> Your intuition and suggestions are appreciated. We put the original structure of the M\"obius strip in Appendix F.3 of the supplementary material, which is indeed a loop as you expected. In the main body, we demonstrate the result in another view: color is drifted in counterclockwise direction on the x-y plane from $\angle xy=0$ to $\angle xy=2\pi$, so the embedding performance is more likely a line. In the final version, we will add the figures in the main body.

---

> > ### Comment · Reviewer_SsqM · 2023-08-17
> >
> > I have read through the reviews and the authors' responses. They have addressed my concerns and I have raised my score accordingly.

---

> > > ### Author Response · Authors · 2023-08-18
> > >
> > > We are very grateful for your re-considering our work. We would keep improving our manuscript to involve your insights.

---

### Author Rebuttal · Authors · 2023-08-09

Dear Program Chairs, Area Chairs, and Reviewers,

First of all, we would like to thank you for your time, constructive critiques, and valuable suggestions, which greatly help us improve the work. We are grateful to reviewers SsqM and sG91 for recognizing the novelty and significance of our work. They acknowledge its potential to have a high impact in addressing asymmetric kernel cases. And we also grateful that reviewers unanimously agree that our qualitative experimental results support the effectiveness of the proposed method. We will address the concern about quantitative experiments as below:

**R0.1 Reviewers SsqM and sG91 think quantitative experiments are needed.**

Thanks for your insightful suggestions. We agree with you that quantitative results can better show the advantages. However, for unsupervised learning, this is always a problem. And thus, specifically to the dimension reduction, there is no well-accepted quantitative criteria. So, the recent papers [1,2] also lack quantitative experiments. Following your suggestions, we have employed four evaluation metrics commonly used for clustering. Given that the original data is clustered in a high-dimensional space, our expectation is that the embedded data will also exhibit clustering patterns after dimension reduction.

We have incorporated quantitative experiments to measure the performance of the proposed method (MagDM) in comparison to two other methods (ME and MME). In the first network, we set the forward flow probability to 0.5 and the backward flow probability to P=0/0.2/1.
We then cluster the low-dimensional embeddings of the three methods using the k-means algorithm with k=3.  To evaluate performance for the clustering results, we consider two internal evaluation metrics: the silhouette coefficient (SC) and the Davies-Bouldin index (DB), as well as two external evaluation metrics: the adjusted Rand index (ARI) and the normalized mutual information (NMI). These metrics assess the quality of clustering. If you are interesting about these indexes, we are willing to discuss with you.

The results of the quantitative experiments evaluating the performance of the proposed MagDM are presented in Table 1 of the attached PDF file. It can be seen that MagDM achieves higher scores in the metrics. Particularly when P=1, where interconnections are more complicated, MagDM significantly outperforms other methods, which highlights the superior effectiveness and robustness of MagDM compared to other methods.

We will ensure to include the experiments in Appendix of the manuscript due to the space limitation. We sincerely look forward to further discussions with the reviewers.

Best wishes,

Anonymous author(s) of Paper6885

**References**

[1] Fanuel M, Alaíz C M, Fernández Á, et al. Magnetic eigenmaps for the visualization of directed networks[J]. Applied and Computational Harmonic Analysis, 2018, 44(1): 189-199.

[2] Gomez A A, Neto A J S, Zubelli J P. Diffusion representation for asymmetric kernels[J]. Applied Numerical Mathematics, 2021, 166: 208-226.

---

### Decision · Program_Chairs · 2023-09-21

**Decision:**

Accept (poster)

**Comment:**

The paper discusses asymmetric kernels for diffusion models. Three reviewers voted to accept the paper, with two significantly raising their scores following the discussion. The use of the magnetic transform was judged to be novel and the quantitative evaluations lent some support for its usefulness. The reviewers commented on the lack of quantitative evaluation.